# Magnetic field expulsion in optically driven YBa$_2$Cu$_3$O$_{6.48}$

S. Fava[1,4], G. De Vecchi[1,4], G. Jotzu[1,4 ✉], M. Buzzi[1,4 ✉], T. Gebert[1], Y. Liu[2], B. Keimer[2] & A. Cavalleri[1,3 ✉]

Coherent optical driving in quantum solids is emerging as a research frontier, with many reports of interesting non-equilibrium quantum phases[1–4] and transient photo-induced functional phenomena such as ferroelectricity[5,6], magnetism[7–10] and superconductivity[11–14]. In high-temperature cuprate superconductors, coherent driving of certain phonon modes has resulted in a transient state with superconducting-like optical properties, observed far above their transition temperature $T_c$ and throughout the pseudogap phase[15–18]. However, questions remain on the microscopic nature of this transient state and how to distinguish it from a non-superconducting state with enhanced carrier mobility. For example, it is not known whether cuprates driven in this fashion exhibit Meissner diamagnetism. Here we examine the time-dependent magnetic field surrounding an optically driven YBa$_2$Cu$_3$O$_{6.48}$ crystal by measuring Faraday rotation in a magneto-optic material placed in the vicinity of the sample. For a constant applied magnetic field and under the same driving conditions that result in superconducting-like optical properties[15–18], a transient diamagnetic response was observed. This response is comparable in size with that expected in an equilibrium type II superconductor of similar shape and size with a volume susceptibility $\chi_v$ of order −0.3. This value is incompatible with a photo-induced increase in mobility without superconductivity. Rather, it underscores the notion of a pseudogap phase in which incipient superconducting correlations are enhanced or synchronized by the drive.

Several recent experiments have made use of ultrashort pulses to dynamically reduce or enhance signatures of superconductivity. For example, irradiation with visible or ultraviolet pulses has been used to study the disruption and recovery of the superconducting state[19–23] (Extended Data Fig. 1, top panels). More recently, mid-infrared optical pulses at 15 µm centre wavelength were used to drive YBa$_2$Cu$_3$O$_{6+x}$ along the insulating $c$-axis direction, coupling to apical oxygen vibrations[24], to coherently drive the material and change its average electronic properties. For this type of excitation, heating under an external drive is minimized[25] and new types of coherence can be activated. A transient state with superconducting-like optical properties[15–17] was observed up to temperatures far above the equilibrium superconducting transition temperature $T_c$ and throughout the equilibrium pseudogap phase. Representative transient optical spectra measured after phonon excitation are reported in Extended Data Fig. 1 (bottom panels).

The specific procedure used to extract the optical properties as well as their significance have been debated in the literature[14,26–36]. Most importantly, these observations are not easily differentiated from those of a transient state with a large increase in mobility. Therefore, complementary probes of superconductivity, to be adapted to the transient nature of this phenomenon, are needed.

In this paper, we search for an ultrafast magnetic field expulsion when the material is excited in a static applied magnetic field. In analogy with field-cooled Meissner diamagnetism, non-superconducting high-mobility carriers would not modify the static magnetic field surrounding the sample, as they would oppose only changes in the magnetic flux. Conversely, a photo-induced state with superconducting correlations would expel the magnetic field from its interior because of the Meissner effect[37].

## Ultrafast optical magnetometry

We adapted the existing techniques of magneto-optical imaging[38] and combined them with magneto-optic sampling of THz pulses[39], to track the ultrafast evolution of magnetic fields surrounding the sample. The Faraday rotation of a probe laser pulse traversing a gallium phosphide (GaP) (100) crystal placed near the excited superconductor, yielded the value of the time- and space-dependent magnetic field with a sensitivity better than 1 µT. Compared with previous measurements of superconductors carried out with ferrimagnetic detectors[40–42], in which the time resolution is limited to about 100 ps, the use of diamagnetic detectors enabled us to follow the magnetic dynamics with a time resolution better than around 1 ps, at the expense of a smaller signal and longer measurement times.

[1]Max Planck Institute for the Structure and Dynamics of Matter, Hamburg, Germany. [2]Max Planck Institute for Solid State Research, Stuttgart, Germany. [3]Department of Physics, Clarendon Laboratory, University of Oxford, Oxford, UK. [4]These authors contributed equally: S. Fava, G. De Vecchi, G. Jotzu, M. Buzzi. ✉e-mail: gregor.jotzu@mpsd.mpg.de; michele.buzzi@mpsd.mpg.de; andrea.cavalleri@mpsd.mpg.de

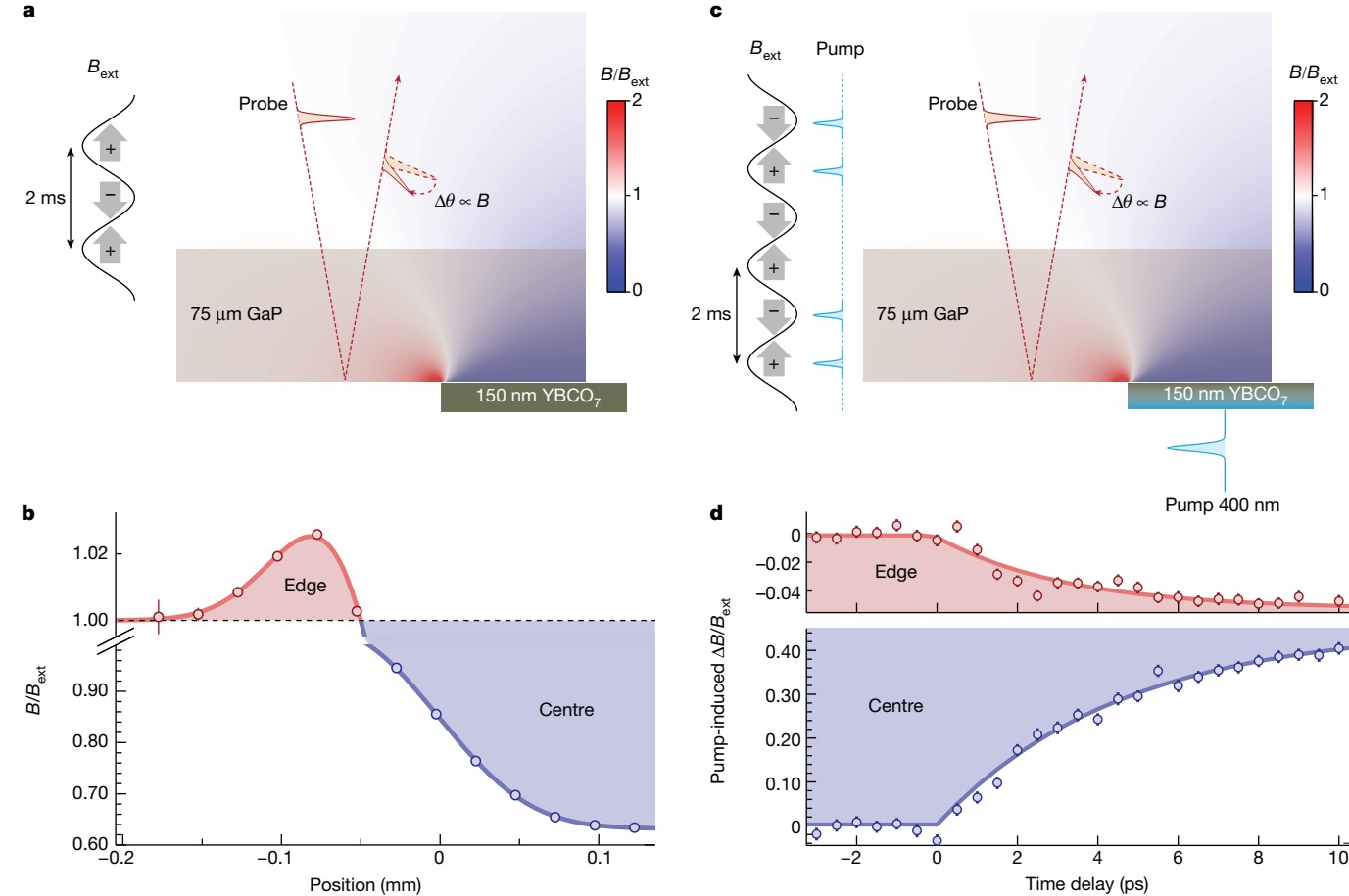

**Fig. 1 | Ultrafast optical magnetometry probing the magnetic response of a superconductor. a**, At equilibrium below $T_c$, a YBCO$_7$ half-disc of 400 µm diameter expels an externally applied magnetic field $B_{ext}$ (see Extended Data Fig. 2 for a full three-dimensional view). The colour plot shows the local changes in the magnetic field calculated assuming a homogeneous $\chi_v \approx -1$ magnetic susceptibility in the superconductor. These changes are measured by tracking the Faraday polarization rotation of a linearly polarized 800-nm probe pulse, reflected after propagation through a GaP (100) crystal placed in close proximity to the sample. To isolate the magnetic contributions to the polarization rotation, the applied magnetic field polarity is periodically cycled (Supplementary Information section 3). **b**, Ratio of the local magnetic field $B$ to the applied $B_{ext}$, measured as a function of distance across the edge of the sample. An increase in the magnetic field is measured near the sample edge with a vacuum (red)

and a reduction above its centre (blue). The data intercept the $B/B_{ext} = 1$ line approximately 0.05 mm away from the YBCO$_7$ sample edge, because of the thickness of the GaP detector, which led to an averaging of the sensed magnetic field in the vertical direction (Extended Data Fig. 3). **c**, The same YBCO$_7$ half-disc shown in Fig. 1a was cooled below $T_c$ and photo-excitation was performed with an ultraviolet laser pulse to disrupt the superconducting state. The time- and spatial-dependent pump-induced changes in the local magnetic field were quantified analogously to Fig. 1a using a double modulation scheme (Supplementary Information section 3). **d**, Pump-induced changes in the local magnetic field $\Delta B$ normalized by the external magnetic field $B_{ext}$, measured near the edge (red) and near the centre of the half-disc (blue) as a function of pump–probe delay. The error bars denote the standard error of the mean.

We first validated the applicability of this technique by measuring the equilibrium superconducting transition in YBa$_2$Cu$_3$O$_7$ (YBCO$_7$) at equilibrium. The experimental configuration is shown in Fig. 1a. The sample was a thin film (150 nm) of YBCO$_7$ grown on Al$_2$O$_3$, out of which a 400-µm diameter half-disc shape with a well-defined edge was created using optical lithography. The YBCO$_7$ film was kept at at a temperature $T = 30$ K $< T_c$ with a spatially homogeneous 2 mT magnetic field applied perpendicular to the plane of the film (vertical direction in Fig. 1a) and generated using a Helmholtz coil pair. An approximately 75-µm thick (100)-oriented GaP crystal was used as magneto-optic detector and was placed directly on top of the sample. A linearly polarized 800-nm, ultrashort probe pulse was focused down to a spot size of about 50 µm on the GaP crystal, impinging at near-normal incidence. The Faraday effect induced a polarization rotation on the beam reflected from the second GaP–vacuum interface, providing a measurement of the vertical component of the local magnetic field, averaged over the probed volume, which was traversed twice because of the reflection geometry. In this experiment, the polarity of the external magnetic field $B_{ext}$ was

periodically cycled, and the signals acquired with $+B_{ext}$ applied field were subtracted from those acquired with $-B_{ext}$. In this way, only contributions to the polarization rotation induced by the local magnetic field were measured (see Extended Data Fig. 2a–c and Supplementary Information sections 2 and 3 for more details on the experimental setup).

Below $T_c$, the superconductor excludes magnetic fields from its interior, and changes in the vertical component of the $B$ field can be estimated using a magnetostatic calculation (see Methods and Extended Data Fig. 3 for more details). The results of this calculation are shown in the colour plot overlaid with the experimental geometry. The magnetic field outside the sample is expected to be reduced in the centre and enhanced near the edge as magnetic flux is excluded from the sample (Fig. 1a, blue and red regions, respectively). These changes were probed by scanning the probe beam in the horizontal direction and by measuring the sensed magnetic field as a function of distance from the edge.

The results of this measurement are shown in Fig. 1b. As predicted, we observed a reduction in the measured magnetic field when measuring above the sample (blue-shaded area) and a corresponding

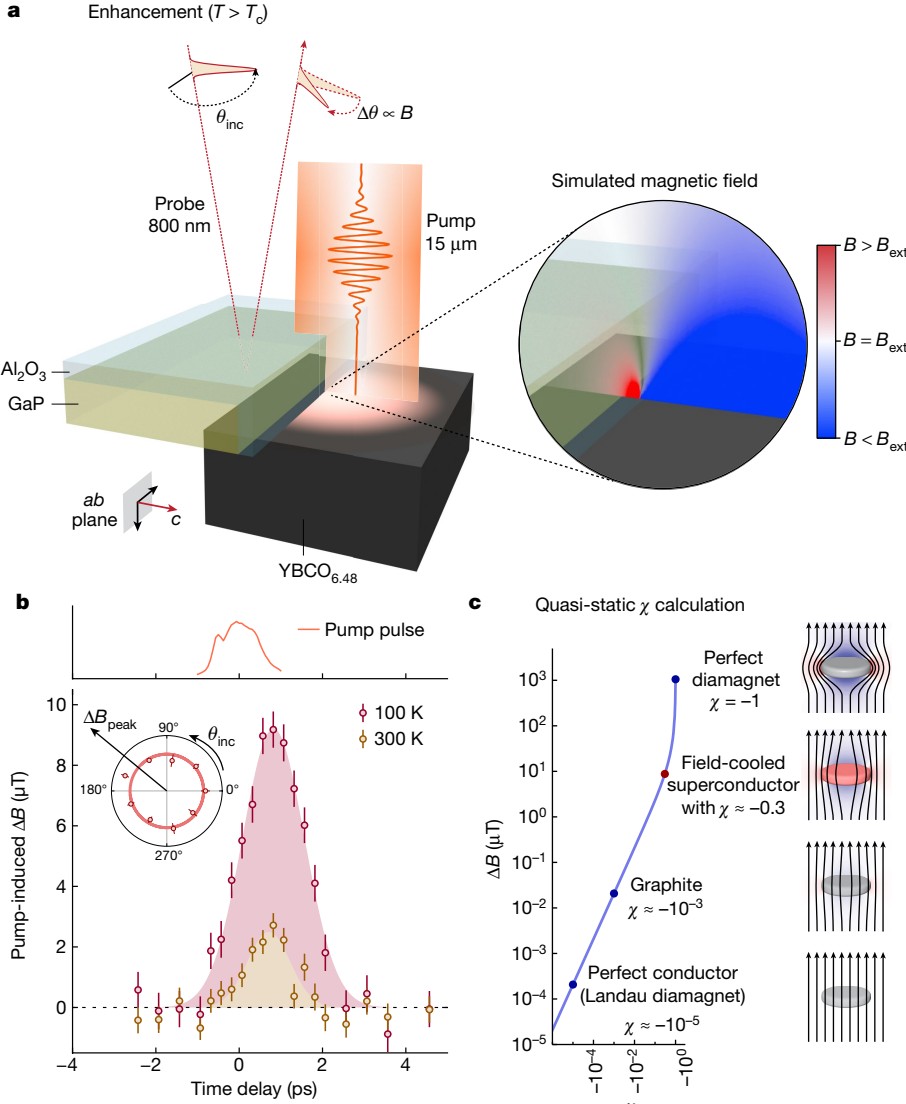

**Fig. 2 | Magnetic field expulsion after phonon excitation in YBCO$_{6.48}$.**
**a**, Schematic of the experiment. A thin Al$_2$O$_3$ crystal is placed on top and next to the exposed side of the GaP (100) detection crystal to completely reflect the 15-μm pump and prevent it from generating a spurious nonlinear optical response in the GaP (100) crystal. The thin Al$_2$O$_3$ crystal also creates a well-defined edge in the mid-infrared pump beam, shaping the photo-excited region into a half-disc of about 375 μm diameter. The expected changes due to the local magnetic field expulsion on photo-excitation are shown in the magnified area on the right. The time-dependent magnetic field is sampled positioning the probe beam in the vicinity of the edge of the photo-excited region. **b**, Pump-induced change in the measured magnetic field ΔB as a

function of pump–probe delay measured at two different temperatures of 100 K (red) and 300 K (yellow). The top plot shows the cross-correlation of the pump and the probe pulses measured in situ, in a position adjacent to the sample. Its peak defines the time zero in the delay scan. The inset shows the dependence of the peak value of the pump-induced magnetic field expulsion ($\Delta B_{peak}$) on the input polarization angle $\theta_{inc}$ of the probe pulse. **c**, Results of a magnetostatic calculation accounting for the geometry and placement of the detector that relate the sensed magnetic field change to a change in the magnetic susceptibility of the photo-excited region in YBCO$_{6.48}$. Further details on this calculation are given in the Methods and Extended Data Figs. 3 and 4. The error bars denote the standard error of the mean.

enhancement near the edge (red-shaded area). The different amplitudes of the effect measured in these two locations are determined by the geometry of the experiment. As shown in the colour plot in Fig. 1a, the decay of the changes in the local magnetic field along the vertical direction is steeper near the edge than above the centre of the sample. Because the magneto-optic detector averaged the magnetic field along the vertical direction, a smaller amplitude signal was detected when measuring near the edge at which the field experienced a faster decay (Methods and Extended Data Fig. 5).

A second test for time-resolved magnetometry was performed in the excitation conditions of Extended Data Fig. 1a. We tracked the dynamics of the magnetic field expulsion after disruption of superconductivity in a YBCO$_7$ thin film by an ultraviolet (400 nm) pulse. The geometry of

the experiment (Fig. 1c) was the same as those used in the equilibrium measurements, with the only addition of the pump pulse, which struck the sample from the side opposite to the detector. Note that the YBCO$_7$ thin film was completely opaque to 400 nm radiation, and the beam was shaped as a half-Gaussian to match the half-disc shape defined in the sample. This geometry ensured that the magneto-optic detector never interacted directly with the optical pump (for more experimental details, see Methods and Extended Data Fig. 6a).

The pump-induced changes in the local magnetic field were measured as a function of pump–probe time delay in two different positions: near the edge and above the sample. The results of these measurements are shown in Fig. 1d. As superconductivity is disrupted (see Extended Data Fig. 1a–c), the magnetic field penetrates into the sample within a

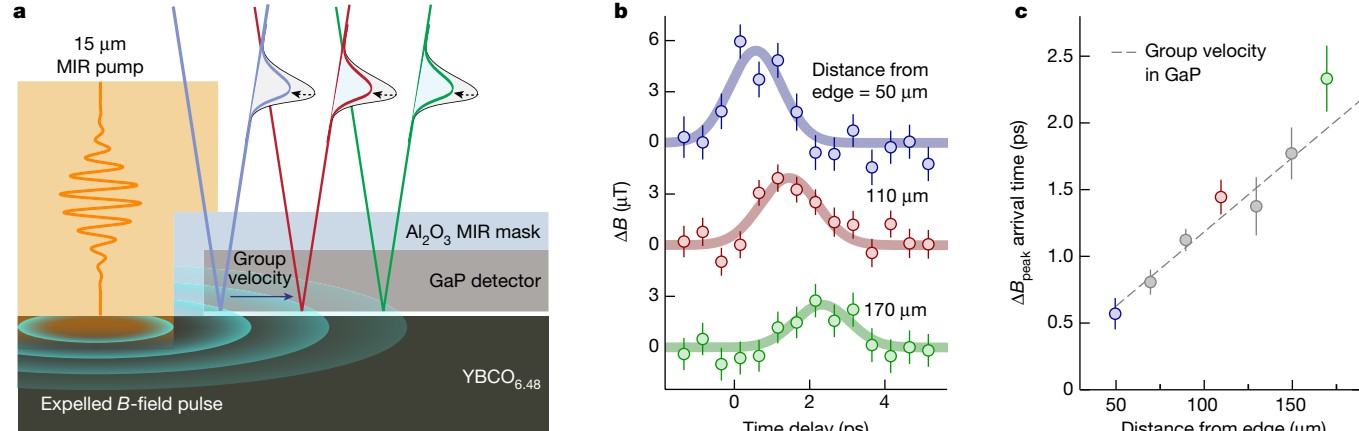

**Fig. 3 | Propagating electromagnetic wave emerging from the photo-excited region. a**, Sectional view of the experimental configuration. Here measurements are performed at a base temperature $T = 100$ K, at different distances from the edge of the photo-excited region. **b**, Pump-induced changes in the local magnetic field measured as a function of pump–probe delay, shown at three selected positions of 50 μm (blue symbols), 110 μm (red symbols) and 170 μm (green symbols). The solid lines are Gaussian fits to the data to extract the peak amplitudes and arrival times. The error bars denote the standard error of the mean. **c**, Magnetic field peak arrival times extracted from Gaussian fits to the time-delay traces measured at different distances from the edge. The grey dashed line shows the expected increase in propagation time with distance based on the group velocity for a 1-THz electromagnetic wave in GaP. The error bars denote the standard error of the centroid parameter extracted from the fits.

few picoseconds, causing a small decrease in the magnetic field near the edge of the sample (red symbols) and a large increase above the sample centre (blue symbols). These changes were measured to persist for several picoseconds. Further details about these measurements and additional data are provided in the Methods and Extended Data Fig. 6b.

## Light-induced magnetic field expulsion

We next turn to the core results of this work and the measurement of magnetic field expulsion in YBa$_2$Cu$_3$O$_{6.48}$ (YBCO$_{6.48}$) after excitation with 15-μm mid-infrared pulses. For the case of photo-induced superconductivity studied here, we expect to observe pump-induced magnetic field changes opposite to the ones measured when disrupting superconductivity and shown in Fig. 1d. Photo-induced enhancement of Meissner screening would rather result in a positive change near the edge and a negative change above the area of photo-excitation.

A YBCO$_{6.48}$ single crystal with an exposed $ac$-surface was held at a series of temperatures $T > T_c$ and irradiated using approximately 1 ps long pulses centred at 15 μm wavelength, with a peak field strength of around 2.5 MV cm$^{-1}$. A homogeneous magnetic field $B_{ext}$, tuned to amplitudes between 0 mT and 12.5 mT, was applied along a direction perpendicular to the sample surface, along the $ab$ planes of YBCO$_{6.48}$. These magnetic field values were lower than the documented[43] $H_{c1}$ in optimally doped YBa$_2$Cu$_3$O$_{6+x}$, and comparable to $H_{c1}$ measured in this sample.

Pump-induced changes to the local magnetic field were measured using the same ultrafast magnetometry technique discussed above. Figure 2a shows the measurement configuration. Owing to the thickness of the single-crystalline $ac$-oriented sample (about 500 μm), being substantially greater than the penetration depth of the mid-infrared light (around 1 μm), both pump and probe were made to impinge onto the sample or detector from the same side. The detector was placed near the edge of the photo-excited region and was completely shielded from the mid-infrared pump by two 30-μm-thick z-cut Al$_2$O$_3$ crystals, which are highly reflective at 15 μm, placed above the detector and on its side (Extended Data Fig. 2d,e). Importantly, the Al$_2$O$_3$ crystals also created a sharp edge for the region of photo-excitation, a prerequisite to maximize the magnetic field changes in its vicinity. Furthermore, the cut of the GaP crystal was chosen to have zero electro-optic coefficient, to avoid confounding contributions to the probe-polarization rotation. In this experiment, the field polarity was periodically cycled, and the pump pulses reached the sample at half the repetition rate of

the probe pulses, yielding double-differential pump–probe measurements. Because the pump-induced changes in the magnetic field measured with applied field $-B_{ext}$ were subtracted from those acquired with applied field $+B_{ext}$, the results reported are independent of possible magnetic fields directly induced by the pump. The time-dependent magnetic field measurements shown here report only changes in the magnetic susceptibility induced by the phonon excitation (Supplementary Information sections 2 and 3 for more details).

Figure 2b shows the pump-induced changes in the local magnetic field measured about 50 μm from the edge created by the mid-infrared (MIR) mask. The measurements were performed as a function of pump–probe time delay at two different base temperatures of 100 K (red symbols) and 300 K (yellow symbols). On photo-excitation, a prompt increase in the magnetic field was observed, peaking at a value of about 10 μT (= $B_{ext}$/1,000) at 100 K and of about 3 μT (= $B_{ext}$/3,000) at 300 K. This transient magnetic field expulsion persisted for about 1 ps, a duration comparable to the lifetime of the superconducting-like conductivity spectra shown in Extended Data Fig. 1f.

A first evaluation of these data is consistent with a photo-induced Meissner response. For the experimental geometry of Fig. 2a, we first compare the measured changes in magnetic field to those expected for a quasi-static change in magnetic susceptibility. As discussed below, this assumption is not entirely justified, although it provides a good estimate of the size of the effect. Assuming that the changes in the photo-excited region are homogeneous, and in the fictitious situation of a slow change in the susceptibility of the material, the raw magnetic field changes would correspond to a quasi-static induced $\chi_{calc}$ of order −0.3, as shown in Fig. 2c. We emphasize that this value is many orders of magnitude greater than what would be observed if the material were made a perfect conductor (assuming weak Landau diamagnetism). Even if it were transformed into a non-superconducting material with the strongest known metallic diamagnetism, such as that observed in graphite planes, the effect would be nearly three orders of magnitude smaller than that observed in this measurement. Rather, the colossal photo-induced diamagnetism observed here is reproduced in equilibrium only by a field-cooled type II superconductor (Methods and Extended Data Fig. 4).

As an extra check, we measured the magnetic field expulsion as a function of the incoming probe-polarization angle $\theta_{inc}$. The polar plot in the inset of Fig. 2b shows the dependence on $\theta_{inc}$ of the measured magnetic field expulsion at the peak of the response. No variations

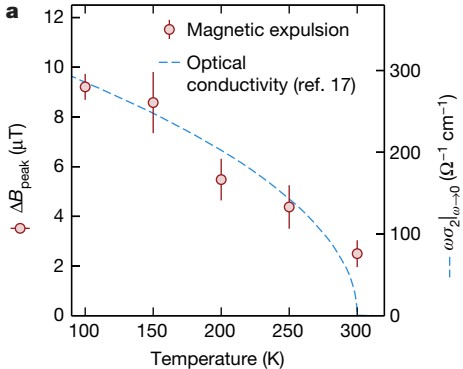

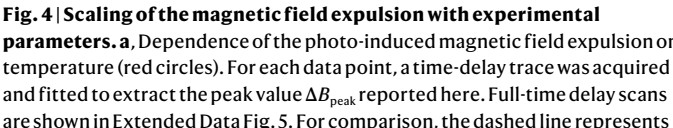

**Fig. 4 | Scaling of the magnetic field expulsion with experimental parameters. a**, Dependence of the photo-induced magnetic field expulsion on temperature (red circles). For each data point, a time-delay trace was acquired and fitted to extract the peak value $\Delta B_{peak}$ reported here. Full-time delay scans are shown in Extended Data Fig. 5. For comparison, the dashed line represents the temperature dependence of the photo-induced superfluid density obtained from optical experiments in ref. 17. **b**, Peak magnetic field expulsion $\Delta B_{peak}$ measured for different applied magnetic fields at a fixed time delay $t = 0.75$ ps and at a base temperature $T = 100$ K. The error bars denote the standard error of the mean.

were observed as $\theta_{inc}$ was rotated between 0 and $2\pi$ corroborating our assignment to a Faraday effect. Other effects, such as the Pockels effect, which should be zero because of the GaP crystal cut, would also show a different dependence on the input polarization (see Supplementary Information section 3 for a detailed discussion and Extended Data Fig. 8 for other verification measurements).

We next discuss the measured signal taking into account that the changes in diamagnetic susceptibility occur dynamically rather than quasi-statically. Because the change in magnetic field inside the material takes place within approximately 1 ps, the photo-excited area of the sample hosts a rapid change in magnetic field ($dB/dt$). Such a time-varying magnetic field is expected to be a source of a picosecond-long propagating electromagnetic pulse (see schematic representation in Fig. 3a). Figure 3b shows pump-induced changes in the local magnetic field measured at three selected distances of 50 μm, 110 μm and 170 μm from the edge of the photo-excited region (along the horizontal direction; Fig. 3a). As the propagation distance was increased, the signal was attenuated and peaked at longer delays, as expected for a propagating electromagnetic wave. As shown in Fig. 3c, from these values we extracted a propagation speed of about $c/n_g$, where $n_g$ is the group index in GaP at about 1 THz frequency[44].

Finally, we show the light-induced magnetic field expulsion, plotted as a function of temperature and applied magnetic field. In Fig. 4a, the peak of the pump-induced magnetic field expulsion is shown as a function of temperature (full-time traces are reported in Extended Data Fig. 7). These follow approximately the same temperature dependence of the photo-induced superfluid density, extracted from transient THz data[17], underscoring a common origin for these two physical observations and a correlation with the temperature scale of the pseudogap. As shown in Fig. 2, the present data show how the mid-infrared drive applied here can generate a colossal diamagnetic response even at room temperature.

Figure 4b shows the same quantity as in Fig. 4a but measured as a function of the applied magnetic field at a fixed temperature $T = 100$ K. Up to the highest $B$ field that could be generated (12.5 mT), the value of the expelled field increased monotonically. Once again, the fact that the effect disappears when $B_{ext} \approx 0$ underscores that the experiment is insensitive to electromagnetic fields carried by the pump. Additional fluence-dependent measurements at a fixed temperature $T = 100$ K are shown in Extended Data Fig. 9.

## Discussion and conclusion

As discussed above, a quench of the magnetic volume susceptibility $\chi_v$ from virtually zero to a value of order −0.3 would give rise to a reduction

in the magnetic field in the photo-irradiated area and an enhancement outside it, explaining the data well.

Alternative explanations could involve the interaction of the drive field with small diamagnetic currents that may already exist before excitation in this material. If we assume that pairing and local superconducting coherence exists throughout the pseudogap phase, it is possible that an amplification mechanism similar to that discussed for Josephson plasma polaritons and reported in ref. 45 could produce a sizeable magnetic field expulsion qualitatively similar to the one observed.

Both of these effects would underscore some form of photo-induced superconductivity. The first mechanism would be compatible with the notion that a true transient superconducting phase is formed. The second mechanism would instead amount to the amplification of pre-existing superconducting fluctuations in the pseudogap phase, hence a truly dynamical phenomenon reminiscent of Floquet superconductivity, loosely defined.

These scenarios highlight the highly unconventional nature of this class of physical phenomena and the part played by coherent electromagnetic fields to engineer quantum materials phases away from equilibrium.

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

## Methods

### Sample preparation and experimental geometries

A $YBCO_7$ thin film grown on two-side polished $Al_2O_3$ was patterned into a half-disc shape using a laser lithography process based on an AZ1512 photoresist mask. After exposure and lift-off, wet etching of the sample was done using a 1% $H_3PO_4$ solution. After etching, the residual photoresist was removed using acetone and isopropanol. Extended Data Fig. 2a shows a micrograph of the $YBCO_7$ film after patterning. The thin films and GaP (100) detector were then mounted onto an $Al_2O_3$ plate that could be fixed directly on the cold finger of the cryostat. $Al_2O_3$ was used to make the sample holder to minimize the effect of eddy currents on the applied field while ensuring good cooling power. A 75-µm thick GaP (100) crystal (SurfaceNet) was used as a detector and put in close contact with the sample (Extended Data Fig. 2b). The detector was polished with a wedge angle of about 1.5° to spatially separate the reflections from the front and back surfaces. This enabled us to detect exclusively the reflection from the back surface, which accumulated the Faraday polarization rotation during its propagation across the detector thickness. Moreover, the GaP back surface and the sample plane were not coplanar, to avoid interference of the reflections from these two surfaces (Extended Data Fig. 2c). The gap between the detector and the patterned $YBCO_7$ film was about 10 µm. This experimental geometry was used for the equilibrium superconductivity and disruption measurements reported in Fig. 1. Further details and calibrations regarding the use of GaP(100) as a magneto-optic detector are provided in Supplementary Information section 4.

The $YBCO_{6.48}$ single crystals were polished after growth to expose an *ac*-oriented surface that allowed access to the crystal *c*-axis. The single crystal sample was glued on the edge of a half-disc-shaped $Al_2O_3$ plate. On the top face of the same $Al_2O_3$ plate, a GaP (100) detector analogous to the one used for the disruption measurements was placed in contact with the $YBCO_{6.48}$ crystal. A 30-µm thick $Al_2O_3$ crystal was placed on top of the GaP crystal and acted as a shield preventing the 15-µm wavelength pump pulses from reaching the GaP detector. A second 30-µm thick $Al_2O_3$ crystal was placed on the side to also protect the detector from the side. Although $Al_2O_3$ is transparent for light at 800 nm wavelength, it is an almost perfect reflector for 15 µm wavelength pulses and features vanishingly small transmission[46] making it the perfect choice for this application. The two thin $Al_2O_3$ crystals also have the function of shaping the pump beam into a half-Gaussian with a well-defined edge. A sketch of this experimental geometry is shown in Extended Data Fig. 2d, alongside a top-view micrograph of the detector and sample assembly in Extended Data Fig. 2e. Also, in this case, the GaP detector was wedged and the $YBCO_{6.48}$ single crystal as well as the $Al_2O_3$ thin crystals were angled to prevent spurious reflections from reaching the polarization analyser (see Supplementary Information section 2 for more details on the experimental setup). Further details on the growth and characterization of these samples are given in Supplementary Information section 1.

### Magnetostatic calculations

The changes in the magnetic field surrounding the sample were calculated in COMSOL using a finite element method to solve Maxwell's equations taking into account the geometry of the experiment. The solution domain was defined as a spherical region of 1 mm radius in which a constant uniform magnetic field was applied. A half-disc-shaped region characterized by a constant, field-independent, spatially homogeneous magnetic susceptibility $\chi_v$ was placed in the centre of the spherical region and was used to model the magnetic response of either the patterned $YBCO_7$ thin film or the photo-excited region in $YBCO_{6.48}$.

Although the size of the half-disc in the simulation was exactly matched to the one used in the experiments for the patterned $YBCO_7$,

assumptions had to be made regarding the size of the photo-excited region in $YBCO_{6.48}$. The latter was modelled as a half-disc of 375 µm diameter, coinciding with the measured 15-µm pump beam spot size, using different thickness values corresponding to different assumptions on the pump penetration depth as discussed below. The weak magnetic response of the substrate or the unperturbed $YBCO_{6.48}$ bulk was not included in the modelling as these are expected to be several orders of magnitude smaller because of their much lower magnetic susceptibility. To account for the detector response that, as written in the main text, generates a polarization rotation that is proportional to the average of the magnetic field in the volume probed by the light pulse, the results of the calculation were integrated along the detector thickness. This yielded two-dimensional maps of the spatially resolved magnetic field that were then convoluted with a two-dimensional Gaussian function to account for the spatial resolution given by the finite size of the focus of the probe beam.

Extended Data Fig. 3a,b shows a comparison between a line scan measured across the straight edge of the $YBCO_7$ half-disc and the results of a magnetostatic calculation performed using geometrical parameters that reflect the experimental conditions. In this simulation, $\chi_v$ was varied to achieve the best agreement with the experimental data. We extracted a value for $\chi_v \approx -1$ that is compatible with the zero-field-cooled magnetic properties of $YBCO_7$ thin film[47].

Extended Data Fig. 3c shows the temperature dependence of the magnetic field measured at equilibrium on top of a $YBCO_{6.48}$ single crystal, using a magneto-optic detector of thickness 250 µm. The geometry of this experiment is analogous to that shown at constant temperature in Fig. 2. At $T_c \approx 55$ K, as the sample turns superconducting, a sudden decrease in the measured magnetic field is observed. Magnetostatic calculations were used to link the measured magnetic field expulsion to the magnetic susceptibility of the $YBCO_{6.48}$ sample. Extended Data Fig. 3d shows a comparison of the extracted magnetic susceptibility $\chi_v$ with that measured on the same sample with a commercial d.c. SQUID magnetometer. The agreement between these two measurements is very good, validating this approach.

Similar calculations were used to quantify the magnetic susceptibility that the photo-excited region in $YBCO_{6.48}$ should acquire after photo-excitation to produce a magnetic field change equal to that measured at the peak of the pump–probe response. This was achieved by running the calculations for a set of $\chi_v$ values and thicknesses of the photo-excited region to obtain calibration curves that related the average magnetic field expulsion measured 50 µm away from the edge to the susceptibility $\chi_v$.

The curve shown in Fig. 2c is calculated under the assumption of a thickness $d$ of the photo-excited region equal to 2 µm, corresponding to the electric field penetration depth of the pump, defined as $d = \frac{c}{\omega \cdot \text{Im}[\tilde{n}_0]}$, where $\tilde{n}_0$ is the stationary complex refractive index of $YBCO_{6.48}$ along the *c*-axis[48] at the pump frequency. This assumption is justified given the sublinear fluence dependence reported in Extended Data Fig. 9.

Extended Data Fig. 4 shows the dependence of the extracted $\chi_v$ on the assumption for the thickness $d$ of the photo-excited region used in the magnetostatic calculations. Three different assumptions are considered:
- $d = 1$ µm, corresponding to the intensity penetration depth of the pump;
- $d = 2$ µm, corresponding to the electric field penetration depth of the pump; and
- $d = 5$ µm, corresponding to the region in which about 99% of the pump energy is absorbed.

We stress that independently of the chosen value for the penetration depth $d$, the retrieved value of $\chi_v$ remains in the $10^{-1}$ range, several orders of magnitude higher than the strongest diamagnetic response observed in metallic systems such as graphite (Fig. 2c).

## Additional spatially resolved equilibrium scans

The data shown in Fig. 1 report spatially dependent scans of the magnetic field surrounding the sample at a single temperature $T = 30$ K, which is less than $T_c$. These measurements were performed in a 2-mT applied magnetic field that switched polarity at 450 Hz frequency, scanning the probe beam across the edge of the YBCO thin film half-disc (Extended Data Fig. 5a). Below $T_c$, an enhancement of the magnetic field close to the edge and a reduction above the sample centre were observed (red and blue symbols in Extended Data Fig. 5b), both associated with a magnetic field exclusion from the sample interior. At $T = 100$ K $> T_c$, no changes in the local magnetic field were observed (grey symbols). This confirms that the presence of a physical edge does not introduce spurious effects (for example, multiple reflections) in our measurement.

This type of measurement was also carried out in the same geometry of Fig. 2 on a bulk YBCO$_{6.48}$ single crystal at a fixed temperature $T = 25$ K $< T_c$ (Extended Data Fig. 5c,d). Similar to the data collected for the YBCO$_7$ thin film, a reduction in the magnetic field above the sample centre and an enhancement near the edge were observed. This confirms that the experimental geometry does not qualitatively affect our observations. In these data, the amount of field enhancement measured near the edge peaks at around 2%, roughly one order of magnitude higher than what was observed in the out-of-equilibrium measurements on the same sample.

This quantitative difference between out-of-equilibrium and equilibrium measurements is well understood based on the following:

1. In the equilibrium measurements, the magnetic field polarity is changed periodically while the sample is in the superconducting state making them effectively analogous to a zero-field-cooled (ZFC) measurement. In the out-of-equilibrium measurements, the field is constant during the photo-excitation of the material, and hence these are the equivalent to a field-cooled type of measurement. ZFC measurements generally give rise to larger signals compared with the field-cooled ones. For example, magnetic susceptibility measurements (Supplementary Information section 1) performed on the same YBCO$_{6.48}$ single crystal show that, even at the lowest temperature, the ZFC susceptibility is about 50 times larger than that of the field-cooled one.

2. The geometry of the two experiments is different. The YBCO$_{6.48}$ sample is a bulk single crystal (2 mm × 0.5 mm and 2 mm thick) and when cooled below $T_c$, the sample becomes superconducting throughout the whole volume. By contrast, in the non-equilibrium case, we expect the magnetic field expulsion to appear only in the photo-excited region, a half-disc (radius of about 150 μm), with a thickness determined by the penetration depth of the pump pulse (around 2 μm). The reduced thickness in the out-of-equilibrium case is expected to give rise to smaller signals (Extended Data Fig. 4) because of different field profiles generated in the vicinity of the sample.

## Spatially resolved pump–probe scans in YBCO$_7$

The data in Fig. 1c,d were acquired in two different positions (above the sample centre and outside of it near the edge) as a function of the time delay between the 800-nm probe pulse and the ultraviolet (400 nm) pump pulse, used to disrupt superconductivity in YBCO$_7$. In Extended Data Fig. 6, we report spatial-dependent measurements of the pump-induced magnetic field changes at one selected time delay $t = 10$ ps. The pump–probe signal shows a spatial dependence similar to that of the static magnetic field exclusion. On the edge of the superconductor, where an enhanced magnetic field is observed at equilibrium, the destruction of superconductivity induced a negative magnetic field change, indicating that the applied magnetic field penetrated back into the sample. By contrast, above the sample centre, a reduced magnetic field is observed at equilibrium and disruption of superconductivity induced a positive signal, indicating that the

magnetic field shielding ceased after photo-excitation. Owing to the specific pulse sequence used for the measurement (Supplementary Information section 3), spatially inhomogeneous trapped magnetic flux was present and the amplitude of the magnetic field change is slightly altered compared with what we would expect from the equilibrium measurements.

## Temperature-dependent delay scans in YBCO$_{6.48}$

A time delay scan, similar to the scans shown in Fig. 2b, was acquired for each temperature. The peak value of each of these scans was extracted by a Gaussian fit of the data, and the peak value was plotted as a function of temperature in Fig. 4a. The data reported in Extended Data Fig. 7 show that the dynamics of the magnetic field expulsion is mostly independent of temperature and only the peak value reduces as the temperature is increased.

## Additional spatial dependences in YBCO$_{6.48}$

Figure 3 shows a probe beam spatial dependence performed by moving the probe at progressively longer distances from the edge of the photo-excited region in the YBCO$_{6.48}$ crystal. In Extended Data Fig. 8, we report additional measurements that were performed at a 50-μm constant distance between the probe and the centre of the excitation beam, moving both of them together from a position where the YBCO$_{6.48}$ crystal is present underneath the GaP detector layer to one where it is not (Extended Data Fig. 8a). The result of this scan is shown in Extended Data Fig. 8b. As both the pump and the probe were moved on the detector away from the sample, the signal vanished confirming that this signal arose from a magnetic field expulsion in the YBCO$_{6.48}$ crystal following photo-excitation and not from spurious interactions in the magneto-optic detection crystal or the Al$_2$O$_3$ filter. This proves that the effect does not originate from the pump pulse itself, rather it is given by photo-excitation of the YBCO$_{6.48}$ sample in an applied magnetic field.

## Fluence-dependent measurements in YBCO$_{6.48}$

In Extended Data Fig. 9, we report the dependence of the measured magnetic field expulsion on the excitation fluence. These data are taken at the peak of the response, at a temperature $T = 100$ K and with an applied magnetic field of 10 mT, after photo-excitation with 15 μm centre wavelength, 1 ps long pulses. The observed scaling seems to be sublinear in fluence. The data in the Figs. 2–4 were acquired at a fluence of around 14 mJ cm$^{-2}$, corresponding in this case to a peak electric field of about 2.5 MV cm$^{-1}$.

## Data availability

Source data are provided with this paper. All other data that support the plots in this paper and other findings of this study are available from the corresponding authors.

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

**Acknowledgements** We acknowledge support from the Deutsche Forschungsgemeinschaft (DFG) by the Cluster of Excellence CUI: Advancing Imaging of Matter (EXC 2056, project ID 390715994). We thank M. Volkmann, P. Licht, I. Khayr and J. Albergoni for their technical assistance. We thank B. Fiedler, B. Höhling and T. Matsuyama for their support in the fabrication of the electronic devices used on the measurement setup; E. König and G. Meier for help with sample fabrication; and J. Harms for assistance with graphics. We also thank P. Lee, E. Demler, M. Michael and D. Basov for their discussions.

**Author contributions** A.C., G.J. and M.B. conceived the experiment. A.C. supervised the project. The setup shown in Fig. 1 was built by G.D.V., G.J. and T.G. and related measurements

were performed by G.D.V. The setup shown in Fig. 2 was built by S.F., M.B. and T.G. and related measurements were performed by S.F. Data analysis was performed by S.F., G.D.V., G.J. and M.B. $YBa_2Cu_3O_{6+x}$ single-crystal samples were provided by Y.L. and B.K. The paper was written by S.F., G.D.V., M.B., G.J. and A.C., with contributions from all co-authors.

**Funding** Open access funding provided by Max Planck Society.

**Competing interests** The authors declare no competing interests.

**Additional information**
**Correspondence and requests for materials** should be addressed to G. Jotzu, M. Buzzi or A. Cavalleri.

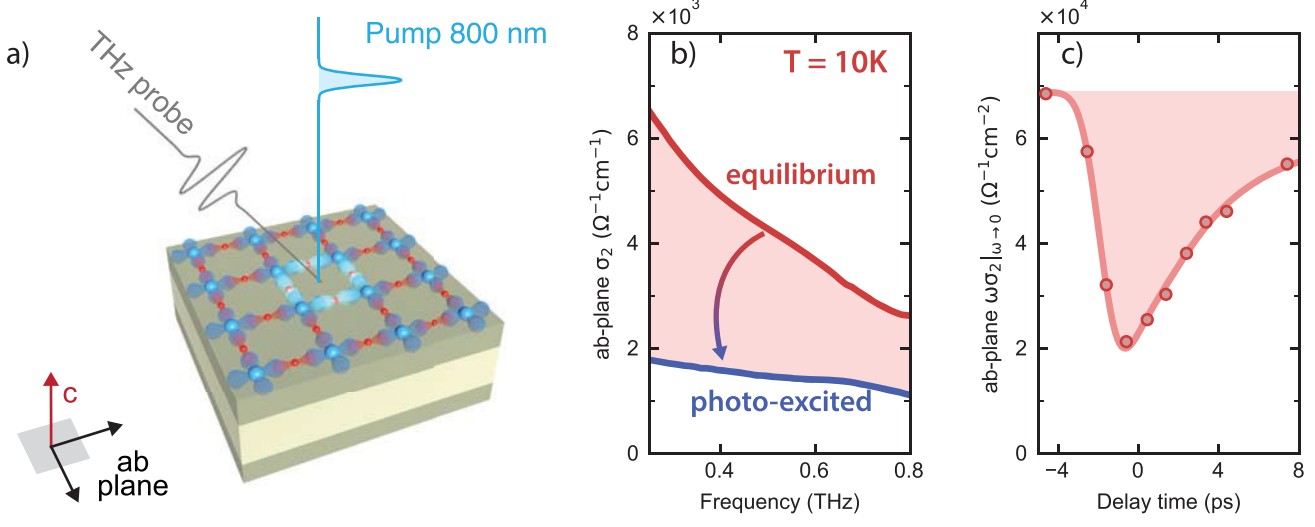

## Disruption (T < T$_c$)

Averitt et al., Phys. Rev. B 63, 140502(R), (2001)

a) THz probe

Pump 800 nm

c

ab plane

b) T = 10K

equilibrium

photo-excited

ab-plane $\sigma_2$ ($\Omega^{-1}$cm$^{-1}$)

Frequency (THz)

c)

ab-plane $\omega\sigma_2|_{\omega\to0}$ ($\Omega^{-1}$cm$^{-2}$)

Delay time (ps)

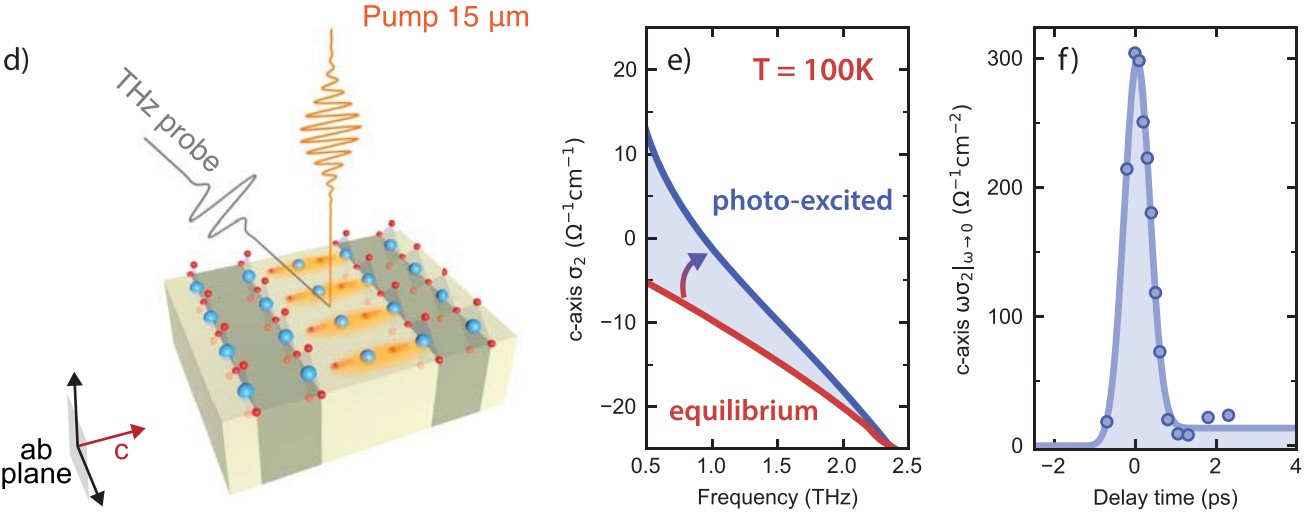

## Enhancement (T > T$_c$)

Liu et al., Phys. Rev X 10, 011053 (2020)

d) THz probe

Pump 15 µm

ab plane

c

e) T = 100K

photo-excited

equilibrium

c-axis $\sigma_2$ ($\Omega^{-1}$cm$^{-1}$)

Frequency (THz)

f)

c-axis $\omega\sigma_2|_{\omega\to0}$ ($\Omega^{-1}$cm$^{-2}$)

Delay time (ps)

**Extended Data Fig. 1 | Transient optical properties of YBa$_2$Cu$_3$O$_{6+x}$ upon photo-excitation.** (**a**) Below $T_c$, femtosecond near-infrared pulses polarized in the ab-plane of YBa$_2$Cu$_3$O$_{6.5}$ are used to break superconducting pairs and disrupt superconductivity. The ab-plane optical conductivity of the transient state is probed with transient THz time domain spectroscopy. (**b**) Imaginary part of the optical conductivity $\sigma_2(\omega)$ measured at equilibrium (red line) and at the peak of the pump-probe response (blue line). At equilibrium $\sigma_2(\omega)$ shows a $1/\omega$ behavior, indicative of dissipationless transport. After photo-excitation, the $1/\omega$ divergence is dramatically reduced. (**c**) Time evolution of $\omega\sigma_2(\omega)|_{\omega\to0}$, a quantity that at equilibrium is indicative of the superfluid density in a superconductor. The time dependence shows a prompt reduction of the cooper pair density after photo-excitation, persisting for several picoseconds. These data are reproduced from ref. 19 (**d**) YBa$_2$Cu$_3$O$_{6.48}$ single crystals are kept at a base temperature $T$ = 100 K > $T_c$ and excited with ~500 fs long, ~19-THz-center-wavelength pulses, resonant with the apical oxygen phonon mode in YBa$_2$Cu$_3$O$_{6.48}$. A time delayed broadband THz pulse probes the c-axis optical conductivity of the sample. (**e**) Same quantity as in (b) but measured along the sample c-axis both at equilibrium (red line) and at the peak of the pump-probe response (blue line). The transient $\sigma_2(\omega)$ spectra show the same -1/$\omega$ behaviour observed along the c-axis in the equilibrium superconducting state[17], suggesting the onset of photo-induced superconducting correlations. (**f**) Same as in (c) but measured along the sample c-axis following mid infrared irradiation. Here, after excitation, a finite transient superfluid density appears. For the experimental conditions explored here, this quantity shows that a superconducting-like response appears and reaches a maximum value within ~1 ps, relaxing back to equilibrium on a similar time scale, compatible with the decay of the driven phonon[45]. In further experiments, these transient optical properties were observed over time windows that ranged from 1 ps to 5 ps, depending on the duration of the drive[18]. Note that ref. 18. reports a far stricter definition of "superconductivity lifetime" taken as the time delay range where the imaginary conductivity diverges as 1/$\omega$ and the transient superfluid density is finite. These data are reproduced from ref. 17.

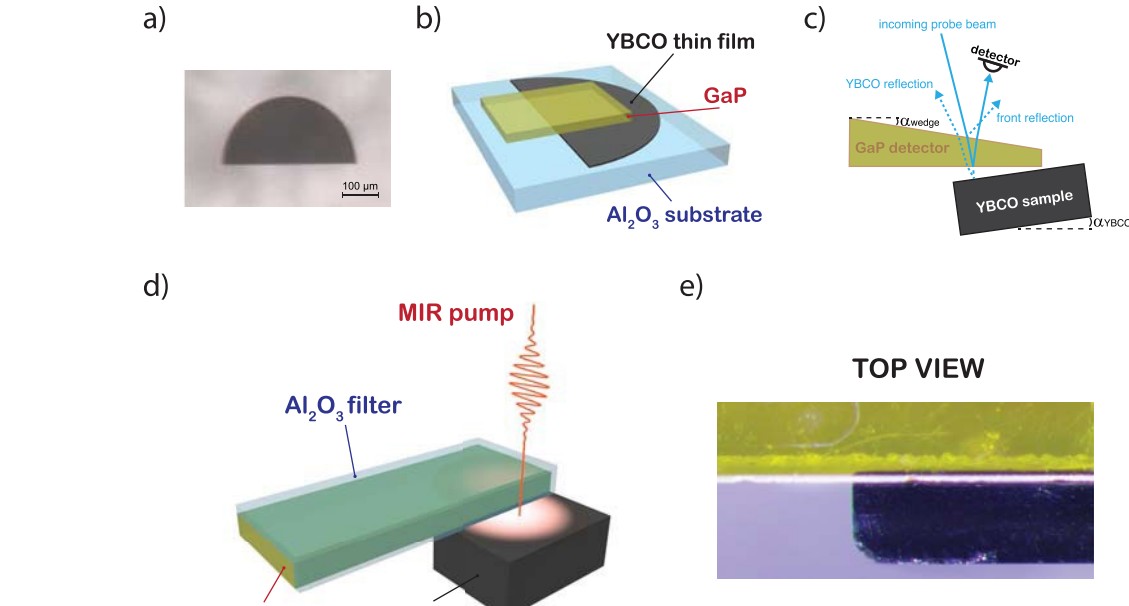

**Extended Data Fig. 2 | Sample preparation and experimental geometries.**
(**a**) Micrograph of the $YBa_2Cu_3O_7$ thin film patterned into a 400 µm diameter half disc shape. (**b**) Sketch of the sample configuration highlighting the positioning of the GaP (100) magneto-optic detector with respect to the $YBa_2Cu_3O_7$ half disc. (**c**) Side view highlighting how undesired reflections are filtered out. The GaP detection crystal is wedged to spatially separate the front and back reflections from each other. The YBCO sample is then mounted with a small tilt with respect to the detector back side. This angle is big enough to filter out the reflection from the YBCO surface, but small enough not to interfere with the measurement. Both angles are exaggerated for clarity, in reality $\alpha_{wedge} \sim 1.5°$ and $\alpha_{YBCO} \sim 1°$. (**d**) Sketch of the sample and detector assembly highlighting the positioning of the $Al_2O_3$ filters and GaP (100) detector with respect to the $YBa_2Cu_3O_{6.48}$ single crystal. (**e**) Top-view micrograph of the sample and detector assembly showing the GaP (100) detector (yellow) seen through the $Al_2O_3$ filter and positioned in the vicinity of the $YBa_2Cu_3O_{6.48}$ single crystal (black). More details on sample preparation and mounting are described in the Methods.

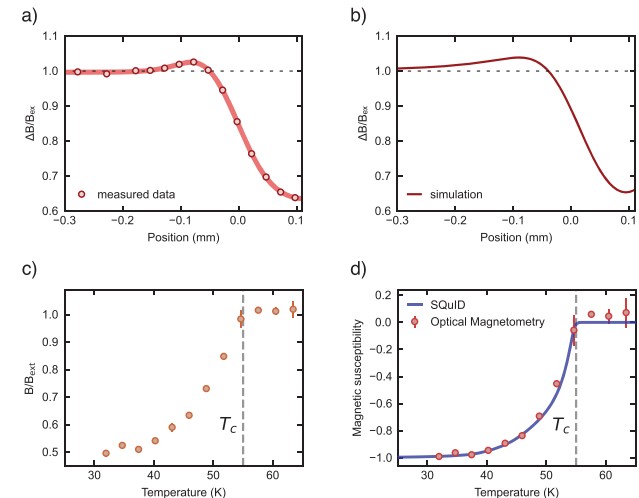

**Extended Data Fig. 3 | Magneto-static simulations. (a)** Ratio of the measured local magnetic field B to the applied one $B_{ext}$ as a function of position across the straight edge of the YBa$_2$Cu$_3$O$_7$ half disc (see Extended Data Fig. 2). The solid line is a guide to the eye. **(b)** Results of a magnetostatic calculation (see Methods) for the set of parameters discussed in the text. Despite the simplicity of the model, the simulated $B/B_{ext}$ shows a good agreement with the experimental data. **(c)** Ratio of the local magnetic field B to the applied one $B_{ext}$, measured as function of temperature on top of the YBaCuO$_{6.48}$ crystal. Upon crossing $T_c \approx 55$ K, a progressive reduction of the measured magnetic field is observed. The applied field was 1 mT and its polarity was switched at 10 Hz frequency. The sampled field did not reach zero even at the lowest temperature because of the finite thickness of the magneto-optic detector, which lead to averaging in the vertical direction. **(d)** Temperature dependent magnetic susceptibility extracted via magnetostatic calculations (filled symbols) from the optical magnetometry data shown in (c). The data is compared to the magnetic susceptibility of the same sample measured with SQuID magnetometry (solid line). The agreement between the data sets is very good.

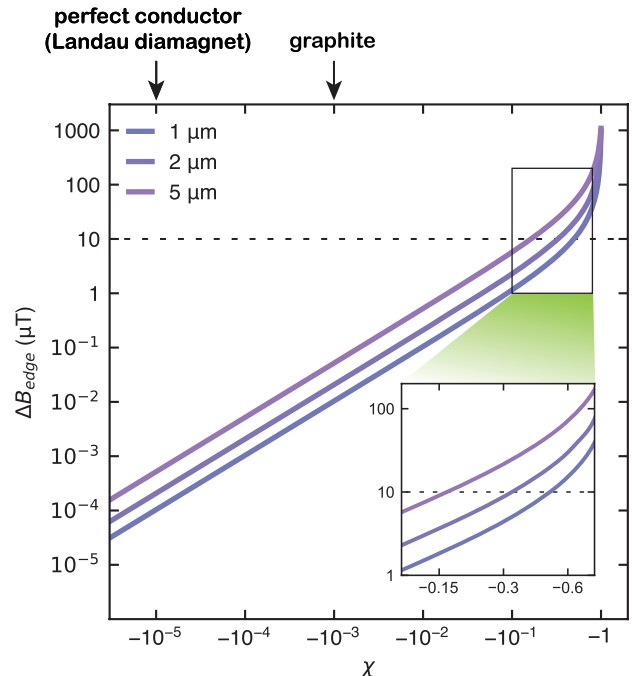

**Extended Data Fig. 4 | Uncertainty estimates on the extracted $\chi_v$ depending on the photo-excited region thickness.** $B$ vs. $\chi_v$ curves obtained for three different values of the thickness of the photo-excited area $d = 1,2,5\,\mu m$. For the geometry shown in Fig. 2c the measured peak field expulsion corresponds to a retrieved value of $\chi_v$ that remains in the $10^{-1}$ range independently on the assumption chosen for the value of $d$. This value is several orders of magnitude higher than the strongest diamagnetic response observed in metallic systems such as graphite (see Fig. 2c and Methods for more details).

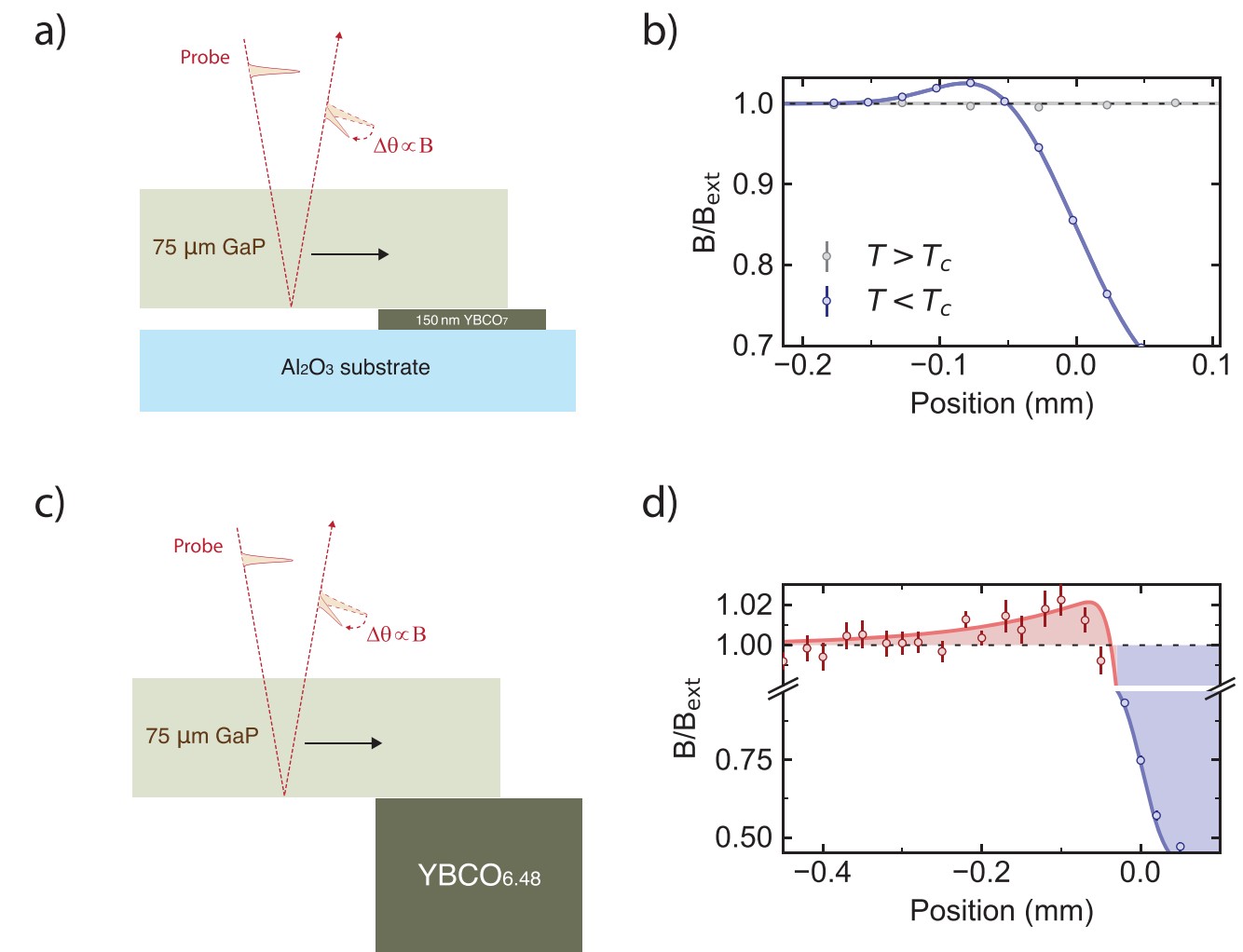

**Extended Data Fig. 5 | Spatially resolved equilibrium scans in YBa₂Cu₃O₆₊ₓ.** (**a**) Sketch of the experimental geometry for the equilibrium measurements on YBa₂Cu₃O₇. A 150 nm thick, 400 μm diameter YBCO half disc (see Extended Data Fig. 2a, b) deposited on an Al₂O₃ substrate is placed in close proximity to the GaP detector. The probe beam is scanned across the edge of the sample to resolve the spatial dependent magnetic field. (**b**) Ratio of the local magnetic field $B$ to the applied one $B_{ext}$, measured as a function of distance from the edge of the YBa₂Cu₃O₇ half-disk at two different temperatures, below $T_c$ (blue) and above $T_c$ (gray). (**c**) Sketch of the experimental geometry for the equilibrium measurements on YBa₂Cu₃O₆.₄₈. A bulk single-crystal sample is placed in close proximity to the GaP detector (Extended Data Fig. 2d,e). The probe beam is scanned across the edge of the sample to resolve the spatial dependent magnetic field. (**d**) Same as in (b) but measured across the edge of the YBa₂Cu₃O₆.₄₈ single crystal. The measurement is performed at $T = 25$ K $<T_c$. The solid line is a guide to the eye.

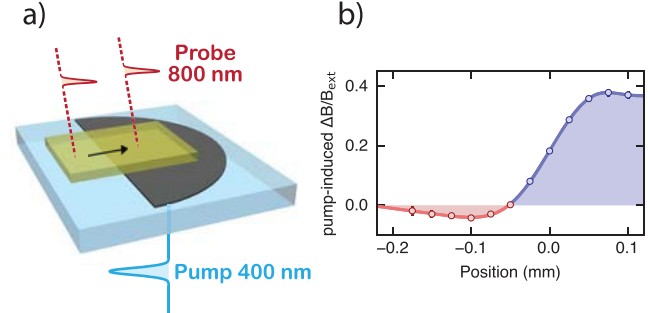

a)

Probe
800 nm

Pump 400 nm

b)

pump-induced $\Delta B/B_{ext}$

**Extended Data Fig. 6 | Spatially resolved pump-probe scans in YBa$_2$Cu$_3$O$_7$.**
(**a**) Sketch of the experimental geometry. The probe beam is moved across the
edge of the YBa$_2$Cu$_3$O$_7$ half-disc shaped film while the pump beam strikes the
sample after transmission through the substrate. (**b**) Space-dependent
pump-induced changes in the ratio of the local magnetic field B to the applied
one measured at $T = 30$ K, with an applied magnetic field of 2 mT and pump
fluence of 5 mJ/cm$^2$. The error bars denote the standard error of the mean.

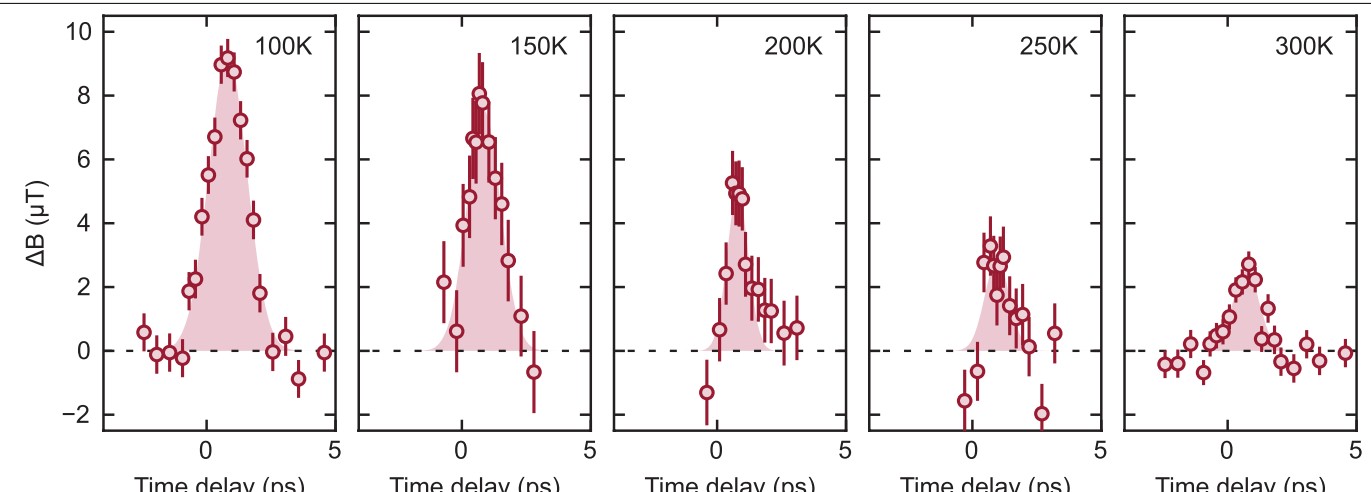

**Extended Data Fig. 7 | Temperature dependent delay scans in YBa$_2$Cu$_3$O$_{6.48}$.** Time dependence of the local magnetic field change near the edge of the YBa$_2$Cu$_3$O$_{6.48}$ crystal measured upon photoexcitation with 15 μm pump pulses at temperatures $T$ of 100 K, 150 K, 200 K, 250 K, and 300 K. The shaded areas display the Gaussian fits used to extract the amplitude of the pump-probe signal. These data were measured at a constant pump peak electric field of 2.5 MV/cm, pump pulse duration of ~1 ps and an applied magnetic field of 10 mT. The error bars denote the standard error of the mean. The data at 100 K and 300 K have been averaged for significantly longer compared to other temperatures, yielding a noticeably lower noise.

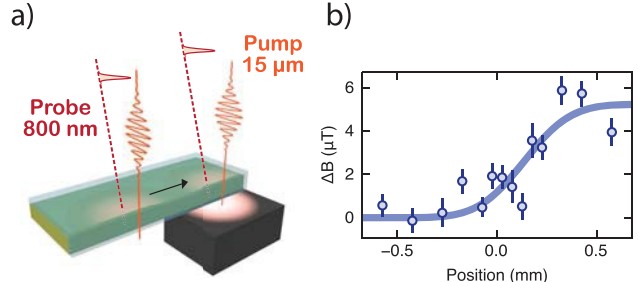

**Extended Data Fig. 8 | Spatially resolved pump-probe scans in YBa₂Cu₃O₆.₄₈.**
(**a**) Sketch of the experimental geometry. The pump and the probe beams are both moved parallel to the edge of the detector from an area where the $YBa_2Cu_3O_{6.48}$ crystal is present underneath the GaP detector layer to one where it is not. (**b**) Measured pump-induced changes in the local magnetic field at a temperature T = 100 K, at a time delay t = 0.75 ps, and peak electric field of 2.5 MV/cm. The solid line is a guide to the eye. The error bars denote the standard error of the mean.

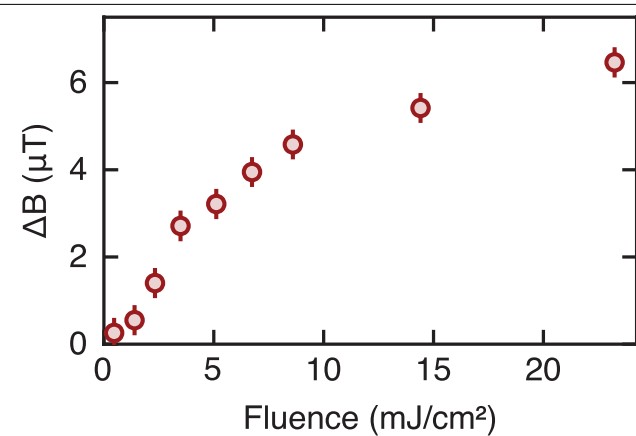

**Extended Data Fig. 9 | Fluence dependence of magnetic field expulsion in YBa$_2$Cu$_3$O$_{6.48}$.** Fluence dependence of the pump-induced changes in the local magnetic field near the edge of the YBa$_2$Cu$_3$O$_{6.48}$ crystal measured upon photoexcitation with 15 μm pump pulses at a temperature of 100 K. These data were measured at the peak of the response and with an applied magnetic field of 10 mT. The error bars denote the standard error of the mean.