## [Peer Review File · Nature]

Manuscript Title: Magnetic field expulsion in optically driven YBa₂Cu₃O_{6.48}

Reviewer Comments & Author Rebuttals

Reviewer Reports on the Initial Version:

Referee #1 (Remarks to the Author):

In the manuscript by Fava et al., the authors presented the observations of transient Meissner effect in light-induced high-T_c cuprates YBCO. The time- and position-dependent magnetic fields around light-driven cuprate samples were measured via Faraday rotation of an optical probe beam in a GaP magneto-optical crystal, which was positioned adjacent to the samples. When light-induced enhancement of superconductivity was created, a reduction (or increase) of magnetic field inside the sample (or immediately outside the sample) was observed, which was attributed to induced Meissner-like diamagnetism.

The results presented in this study are important. Light-induced superconductivity remains debated and controversial, with previous evidence relying mostly on the observation of a $1/\omega$ divergence in the imaginary part of THz conductivity. This study may provide additional evidence for the validation of light-induced enhancement of superconductivity.

However, on the other hand, I do not think the measurements presented in this study are novel. Meissner effect and magnetic field expulsion in superconductors (both equilibrium and nonequilibrium) by magneto-optical effect and Faraday rotation have been studied previously by several groups as demonstrated in Refs [1,2,3]. For example, Refs [1,2] used very similar geometry to measure the magnetic field expulsion for the sample kind of cuprate samples by cooling it down to below T_c, based on similar magneto-optic Faraday rotation method. Ref [3] demonstrated transient Meissner effect of light-induced quenching of superconductivity, using the same method. Compared with literature, the additional insight from this manuscript is that it demonstrated transient Meissner effect of light-induced enhancement (instead of quenching) of superconductivity.

Therefore, in my opinion, I am sorry to say that while this study is very interesting and important, the results are not novel and significant enough to justify publication in Nature. Rather, I believe it could be published in Nature Physics.

Below I have a couple of minor suggestions which need to be addressed before the paper may be published elsewhere:

1. In Fig. 2, based on my understanding, is the left edge of the YBCO sample in (a) aligned with approximately -0.05 mm position in (b), e.g., at the crossing of the curve and the dashed horizontal line? It would be better to indicate the position of the YBCO sample in (b).
2. It would be more effective to redesign the illustration of the sample and detector assembly in Fig. 4(a). Without the help of Fig. S2.2, it is difficult to understand the structure.

References

- [1] Y. Yuan et al., *J. Magnetism and Magnetic Materials* 95, 58-60 (1991).
[https://doi.org/10.1016/0304-8853\(91\)90214-U](https://doi.org/10.1016/0304-8853(91)90214-U)
- [2] M. R. Koblischka et al., *Supercond. Sci. Technol.* 8 199 (1995). <https://doi.org/10.1088/0953-2048/8/4/002>
- [3] M. R. Freeman, *MRS Online Proceedings Library* 290, 323 (1992). <https://doi.org/10.1557/PROC-290-323>

Referee #2 (Remarks to the Author):

Summary of the key results :

In the past decade, the phenomenon of light-induced superconductivity in cuprates has been a contentious and widely discussed subject within the fields of ultrafast spectroscopy and superconductivity. This paper aims to detect possible photoinduced diamagnetism and differentiate the light-induced superconductivity-like state from the high mobility photocarriers.

Interest to the broad physics community :

Should the results prove to be accurate and valid, they will be of general interest to the condensed matter physics community beyond those directly working on ultrafast phenomena in quantum materials.

Originality and significance:

The novel method of detecting light-induced transient changes in the magnetic field of cuprates represents a significant advancement. Through the measurement of Faraday rotation in a GaP (100) magneto-optic crystal under 15 μm mid-infrared pulse excitation, the authors observed a heightened magnetic field at the sample's edge, which they attributed to induced diamagnetism. The original data presented in the manuscript hold the potential for substantial impact if the claim proves to be valid.

Comments and questions:

However, I have serious concerns about the validity and consistency with their earlier observations and claims.

1. Before delving into experimental details, I was perplexed by the claimed pump-probe time window of transient superconductivity in this and previous studies from the group. A notable distinction was observed between the light-induced diamagnetism/superconductivity in the current study and the findings of previous research. The initial influential Science paper by this group (Fausti et al. *Science* 331, 189 (2011), cited as ref. [13]) reported light-induced superconductivity in a striped ordered cuprate LESCO1/8 with a 15 μm wavelength MIR pulse polarized within the ab-plane, where the induced condensate reached a maximum at 5 ps after excitation. They emphasized that the light induced superconductivity lasted over 20 ps or over 100 ps. They also alerted "Photoexcitation with the electric field polarized orthogonal to the planes resulted in only a small reflectivity change during

the pump pulse and no long-lived response” in the paper. Subsequent studies by the same group, including some referenced in this manuscript, utilized intense MIR or NIR pulse excitations with pulse durations ranging from ~100 to 300 fs polarized along c-axis on YBCO and LBCO, and claimed transient superconductivity at 0.8 – 2 ps after excitation. In contrast, the present study involved chirping the 150 fs MIR pulse to a duration of ~1 ps by using a 10 mm NaCl rod, and observed the pump-induced magnetic signal only during the driving period of the MIR pump-pulse. This represents a departure from their prior work in the pump-probe time-delay. The current study appears to challenge or refute the possibility of light-induced superconductivity in their previous research as no effect is induced after pulse duration. To substantiate the reported light-induced phenomenon as superconductivity, it is imperative for the authors to replicate the pump conditions in prior research and observe transient diamagnetism within the same time-delay window, that is, 0.8 – 2 in YBCO/LBCO or over 20 ps in LESCO1/8 after pumping.

2. In the introduction and Figure 1, the authors illustrate that an 800 nm NIR pulse polarized within the ab-plane suppresses superconductivity below T_c , while an MIR pulse along the c-axis induces transient superconductivity above T_c . The authors seem to imply that the NIR pulse may not induce transient superconductivity above T_c , and the MIR pulse may not suppress superconductivity below T_c . However, their previous research on LBCO (e.g., PRB 90, 100503R (2014)) showed that NIR excitation at 800 nm polarized along c-axis not only enhanced superconductivity below T_c but also induced transient superconductivity above T_c . In their research on YBCO (ref.19), the authors also showed that, in addition to the MIR pulse, NIR excitation can induce transient superconductivity. Specifically, in ref. 19, the authors found that an 800 nm NIR pulse on YBCO actually produced the largest condensate $\omega\sigma(2)|_{\omega\rightarrow 0}$, and a 400 nm pulse also induced sizeable $\omega\sigma(2)|_{\omega\rightarrow 0}$. In the current study, the authors only demonstrate the effects of the 400 nm pump under the condition of $T < T_c$ and the 15 μ m pump under the condition of $T > T_c$, as presented in Figures 3 and 4-6, respectively. Such comparisons are inappropriate. To validate their previous findings and ensure consistency, the authors should include the effects of both the NIR (or 400 nm) pump and MIR pump below and above T_c . To address the apparent controversies in different studies, they should also check the dependence of transient signal on pump pulse polarization, if possible.

Following are comments and concerns on the measurement in this work:

1) In the field of ultrahigh-power pulse laser technology, it is well established that irradiating a solid metallic target with a high-power pulse laser can generate a very high pulse magnetic field. This phenomenon is attributed to the generation and expelling of fast electrons from the laser focus spot due to the intense pulse laser driving, resulting in a high electrostatic potential near the focus and charging of the target. The potential then drags background cold electrons to the focus spot, leading to a current that generates a strong pulse magnetic field. Previous studies have demonstrated that a single turn coil with an open-ended plate target can produce a magnetic field of several hundred tesla or higher near the coil within the pulse laser driving period on target. See, for example, Zhu et al. APL 107, 261903 (2015), APL 113, 072405 (2018); Sederberg et al. PRX 10, 011063 (2020).) It is important to note that the induced electron current passing through the straight sample segments connecting the target can cause a pulse magnetic field even around the straight sample segments. In the present study, the mid-infrared (MIR) laser pulse has a peak field of approximately 2.5 MV/cm,

raising the question of whether this could result in a magnetic field change of 10 uT during the pulse driving period. This effect has to be carefully considered.

2) The measurement may be affected by an additional complexity, as the MIR pump pulse itself contains a magnetic field component exceeding 0.8 T. Given that the lifetime of this transient signal closely matches the pump pulse duration, the authors must eliminate the possibility that the detected transient signal arises from the magnetic field component of the pump pulse propagating in the sample. 10 uT is several orders lower than the peak magnetic field of MIR pulse. Is a tiny misalignment of pump pulse can affect the measurement? Carefully designed experiments to investigate the interference of the pump pulse's magnetic field component on the experimental results seem to be necessary.

3) In the magneto-optic sampling of GaP (100) below T_c , a reduction in polarization rotation at the film's center and an increase near its edge were observed. The authors suggested that this change in polarization rotation is linked to a change in the magnetic field. However, scanning the probing pulse from the edge to the center not only alters the magnetic field experienced by the GaP layer but also changes the device's geometry. At the sample's edge, the probing light is reflected by the air-GaP-air multilayers, while at the center, the air-GaP-YBCO multilayer reflects the probing pulse. These different interfaces add complexity to extract a pure magnetic repulsion signal from the total recorded signal, and the authors should discuss possible additional effect from the GaP-YBCO interface. Furthermore, a temperature-dependent measurement of the polarization rotation at the center and around the edge of the sample is obviously necessary. If the recorded polarization rotation signal inherently reflects the Meissner effect, the authors should observe a sudden change in the signal when the temperature is cooled below T_c , similar to the observation in the magnetic susceptibility measurement.

4) In the context of *c*-axis measurements, it is important to address the lack of a spatial dependence scan at equilibrium, a procedure undertaken for the *ab*-plane device. Furthermore, conducting a temperature dependent scan at a specific distance would help clarify the connection between the recorded pump-probe signal and the superconductivity-induced Meissner effect.

In summary, the ultrafast magneto-optic sampling method presented in this study could facilitate further investigation of the previously reported light-induced superconductivity. To validate this phenomenon, it is essential for the authors to reproduce the pump conditions used in earlier research and observe transient diamagnetism within the same time-delay window. Additionally, rigorous efforts are required to ascertain the intrinsic connection between the recorded pump-probe signal and the transient Meissner effect. All the above concerns should be adequately addressed. Lastly, we recommend that the authors include pertinent references in their paper to address the ongoing debate surrounding the light-induced superconductivity-like response in optical pump THz probe measurements.

Referee #3 (Remarks to the Author):

The manuscript "Magnetic field expulsion in optically driven YBCO" by S. Fava et al. deals with the subject of "transient superconductivity", an issue which has attracted significant attention over the past few years. As is clearly outlined in the introduction, a number of experiments has revealed the emergence of new types of coherence when underdoped YBCO is driven along the c-direction with MIR optical pulses. In the induced non-equilibrium state the (imaginary part of the) optical conductivity shows a $\sigma_2 \sim 1/\omega$ behavior, compatible with a zero frequency pole in $\sigma_1(\omega)$ which is the hallmark of superconductivity. The idea is that the optical pulses coherently excite apical oxygen phonons that induce the superconducting-like behavior.

The question, if one can really deduce a transient superconducting state from the optical data has stimulated a lot of controversy. This concerns the measurement process itself (cf. e.g. J. S. Dodge, PRL 130, 146002 (2023)) but also the question if an "equilibrium quantity" as the superfluid stiffness, is a signature of superconductivity if this quantity is formally extracted from $\omega\sigma_2(\omega)$ in a non-equilibrium state. In this regard, I consider the work presented in this manuscript as a significant step forward since authors take the Meissner effect as a proxy for superconductivity, i.e. the (complete) expulsion of a static magnetic field from the interior of a superconductor. In the experimental setup the magnetic field is measured via the polarization rotation by the Faraday effect of a linearly polarized probe pulse, realized by propagation through a GaP crystal in proximity to the sample. This is first demonstrated for a superconductor in equilibrium where authors observe a significant reduction of the field above the sample and, due to the expulsion, an increased magnetic field at the edge of the sample. It is also demonstrated that the opposite effect is observed when the SC state is disrupted by an ultraviolet laser pulse.

The central experiment discussed in the manuscript is sketched in Fig. 4 and concerns the measurement of the magnetic field expulsion for an underdoped YBCO sample which is excited along the c-direction and brought into a transient state where $\sigma_2(\omega)$ indicates possible induced SC properties. It is also shown that the signal is independent of the probe polarization angle and therefore originates from a Faraday and not e.g. the Pockels effect.

Moreover, it is demonstrated that the temperature dependence of the magnetic field expulsion and the non-equilibrium "superfluid density" are close to each other suggesting a common origin of both effects. The paper closes with a discussion of two possible explanations for the observed diamagnetic signal, either due to a real transient SC phase in part of the sample or the amplification of pre-existing diamagnetic currents in the pseudogap phase.

I consider this as a very interesting paper with a novel approach which advances our understanding of the transient state of coherently driven superconductors. The paper is well written and appendices provide sufficient supplementary material so that one can understand the experimental setup and the evaluation of the data.

I have only one major issue which concerns the consistency of the results related to the discussion of Figs. 4,5. The evidence for the conclusion would be significantly strengthened if the authors also gave the equilibrium data below T_c for this sample (i.e. similar to what they discuss in Fig. 2) which would also help to estimate the non-equilibrium data above T_c . In the present version this is accomplished via a quasistatic calculation, which for the equilibrium case would be also more

reliable. Moreover, in previous work of the authors it has been shown that also below T_c , $\omega\sigma_2(\omega)$ is enhanced by the coherent excitation. If the interpretation of the authors is consistent this should also induce a concomitant enhanced magnetic field expulsion.

Provided, authors provide this additional material, I recommend publication of the manuscript in Nature Materials.

Detailed Reply To Referees - Nature 2023-12-22464

In this document comments from the Referees are included in **black**. Author responses are displayed in **blue**.

Referee #1

In the manuscript by Fava et al., the authors presented the observations of transient Meissner effect in light-induced high-T_c cuprates YBCO. The time- and position-dependent magnetic fields around light-driven cuprate samples were measured via Faraday rotation of an optical probe beam in a GaP magneto-optical crystal, which was positioned adjacent to the samples. When light-induced enhancement of superconductivity was created, a reduction (or increase) of magnetic field inside the sample (or immediately outside the sample) was observed, which was attributed to induced Meissner-like diamagnetism.

The results presented in this study are important. *Light-induced superconductivity remains debated and controversial, with previous evidence relying mostly on the observation of a 1/omega divergence in the imaginary part of THz conductivity. This study may provide additional evidence for the validation of light-induced enhancement of superconductivity.*

We thank the Referee for their review. In the statement above, our results are framed in the right context. Indeed, measurements of optically driven YBa₂Cu₃O_{6+x} have so far been shown to exhibit only one of the features of superconductivity, which is connected to the divergent imaginary conductivity. However, this observation does not differentiate between perfect conductivity and superconductivity. Ultrafast magnetometry shows that the induced diamagnetism is very large and is indicative of transient superconductivity.

However, on the other hand, I do not think the measurements presented in this study are novel. *Meissner effect and magnetic field expulsion in superconductors (both equilibrium and nonequilibrium) by magneto-optical effect and Faraday rotation have been studied previously by several groups as demonstrated in Refs [1,2,3]. For example, Refs [1,2] used very similar geometry to measure the magnetic field expulsion for the sample kind of cuprate samples by cooling it down to below T_c, based on similar magneto-optic Faraday rotation method. Ref [3] demonstrated transient Meissner effect of light-induced quenching of superconductivity, using the same method.*

References

- [1] Y. Yuan et al., *J. Magnetism and Magnetic Materials* 95, 58-60 (1991).
[https://doi.org/10.1016/0304-8853\(91\)90214-U](https://doi.org/10.1016/0304-8853(91)90214-U)
- [2] M. R. Koblishka et al., *Supercond. Sci. Technol.* 8 199 (1995).
<https://doi.org/10.1088/0953-2048/8/4/002>
- [3] M. R. Freeman, *MRS Online Proceedings Library* 290, 323 (1992).
<https://doi.org/10.1557/PROC-290-323>

Indeed, our measurements have been inspired by previous applications of ultrafast magnetometry, which are referenced in the manuscript (we have included the additional references suggested by the reviewer in the revised version of the paper). **However, nowhere in the manuscript have we claimed that this technique is used here for the first time.** The only thing to note in this respect is that we have demonstrated magnetometry using a fast diamagnetic detector (see e.g. Riordan et al., *Optical and Quantum Electronics* 32, 489 (2000)), which has sub-picosecond time resolution. This has never been done before. All previous studies used ferro- or ferrimagnetic detectors, with time resolution limited to ~ 100 ps. Improvement of this technique is nevertheless not the key advance reported in our paper.

As recognized by this Referee and amplified by Referees 2 and 3, the main discovery reported here is that of a colossal (Meissner-like) magnetic field expulsion in the driven state of $\text{YBa}_2\text{Cu}_3\text{O}_{6.48}$. This driven state is seen far above T_c and up to room temperature. These observations will in our view be of great interest to the scientific community. The present experiments also set a new benchmark for any theory of non-equilibrium superconductivity in YBCO.

*Compared with literature, the additional insight from this manuscript is that it demonstrated transient Meissner effect of light-induced enhancement (instead of quenching) of superconductivity. Therefore, in my opinion, I am sorry to say **that while this study is very interesting and important, the results are not novel and significant enough to justify publication in Nature.** Rather, I believe it could be published in Nature Physics.*

We respectfully disagree. In our view, the demonstration of light enhancement of superconductivity, is far more interesting, and certainly timely, than disruption.

Below I have a couple of minor suggestions which need to be addressed before the paper may be published elsewhere:

1. In Fig. 2, based on my understanding, is the left edge of the YBCO sample in (a) aligned with approximately -0.05 mm position in (b), e.g., at the crossing of the curve and the dashed horizontal line? It would be better to indicate the position of the YBCO sample in (b).

This is a valid point which was insufficiently clear in the previous version of the manuscript. The edge of YBCO sample in Fig. 2(a) was aligned with the zero in the plot in Fig. 2(b). The measured curve crosses the $B/B_{\text{ext}} = 1$ line approximately 0.05mm away from the YBCO sample edge, due to the finite thickness of the GaP detector, which led to an averaging of the sensed magnetic field in the vertical direction. This effect is detailed in Supplementary Section S6. Due to this averaging, the $B/B_{\text{ext}} = 1$ crossing point is shifted further outside of the sample. To clarify this point we have added an additional explanation in the figure caption.

2. It would be more effective to redesign the illustration of the sample and detector assembly in Fig. 4(a). Without the help of Fig. S2.2, it is difficult to understand the structure.

We thank the Referee for this suggestion, we have simplified the graphic in Figure 4. The new version of figure 4 is reported below in figure R1.1

Figure R1.1 (updated version of figure 4 in the Main Text) Magnetic field expulsion after phonon excitation in $\text{YBa}_2\text{Cu}_3\text{O}_{6.48}$. **(a)** Schematic of the experiment. A thin Al_2O_3 crystal is placed on top and next to the exposed side of the GaP (100) detection crystal to completely reflect the $15\ \mu\text{m}$ pump and prevent it from generating a spurious non-linear optical response in the GaP (100) crystal. The thin Al_2O_3 crystal also creates a well-defined edge in the mid infrared pump beam, shaping the photo-excited region into a half disc of $\sim 375\ \mu\text{m}$ diameter. The expected changes in the magnetic field due to the superconducting transition are shown in the magnified area on the right. Red (blue) changes correspond to an enhancement (reduction) of the local magnetic field on the edge (above the center) of the photo-excited region. The time dependent changes in the local magnetic field expulsion are probed positioning the probe beam in the vicinity of the edge of the photo-excited region (red region in the magnified view). **(b)** Pump-induced change in the measured magnetic field (ΔB) as function of pump-probe delay measured at two different temperatures of 100 K (red) and 300 K (yellow). The upper plot shows the cross-correlation of the pump and the probe pulses measured in-situ in a position adjacent to the sample. Its peak defines the time-zero in the delay scan. The inset shows the dependence of the peak value of the pump-induced magnetic field expulsion (ΔB_{peak}), measured via polarization rotation, on the input polarization angle θ_{inc} chosen for the probe pulse. **(c)** Results of a magnetostatic calculation accounting for the geometry and placement of the detector that relate the sensed magnetic field change to a change in the magnetic susceptibility of the photo-excited region in $\text{YBa}_2\text{Cu}_3\text{O}_{6.48}$. Details of this calculation are contained in Supplementary Section S6. The error bars denote the standard error on the mean.

Referee #2

Summary of the key results :

In the past decade, the phenomenon of light-induced superconductivity in cuprates has been a contentious and widely discussed subject within the fields of ultrafast spectroscopy and superconductivity. This paper aims to detect possible photoinduced diamagnetism and differentiate the light-induced superconductivity-like state from the high mobility photocarriers.

Interest to the broad physics community :

*Should the results prove to be accurate and valid, **they will be of general interest to the condensed matter physics community beyond those directly working on ultrafast phenomena in quantum materials.***

Originality and significance:

The novel method of detecting light-induced transient changes in the magnetic field of cuprates represents a significant advancement.** Through the measurement of Faraday rotation in a GaP (100) magneto-optic crystal under 15 μm mid-infrared pulse excitation, the authors observed a heightened magnetic field at the sample's edge, which they attributed to induced diamagnetism. **The original data presented in the manuscript hold the potential for substantial impact if the claim proves to be valid.

We thank the Referee for the statements above, which complement those of Referee 1 and support our case for scientific significance. These endorsements are nevertheless accompanied by concerns about interpretation, which we address in full below.

Comments and questions:

However, I have serious concerns about the validity and consistency with their earlier observations and claims.

1. Before delving into experimental details, I was perplexed by the claimed pump-probe time window of transient superconductivity in this and previous studies from the group.

A notable distinction was observed between the light-induced diamagnetism/superconductivity in the current study and the findings of previous research. The initial influential Science paper by this group (Fausti et al. Science 331, 189 (2011), cited as ref. [13]) reported light-induced superconductivity in a striped ordered cuprate LESCO1/8 with a 15 μm wavelength MIR pulse polarized within the ab -plane, where the induced condensate reached a maximum at 5 ps after excitation. They emphasized that the light induced superconductivity lasted over 20 ps or over 100 ps. They also alerted "Photoexcitation with the electric field polarized orthogonal to the planes resulted in only a small reflectivity change during the pump pulse and no long-lived response" in the paper.

Subsequent studies by the same group, including some referenced in this manuscript, utilized intense MIR or NIR pulse excitations with pulse durations ranging from ~ 100 to 300 fs polarized along c -axis on YBCO and LBCO, and claimed transient superconductivity at 0.8 – 2 ps after excitation

The Referee highlights a number of quantitative differences between different experiments performed in different materials, excited with different wavelengths and different polarizations. We appreciate that the diverse phenomenology may be cause of some confusion, but quantitative consistency amongst all these is not expected.

Although we have addressed all these comparisons, we really do not think these are a problem and we feel that the Referee will agree with us once they consider our response.

For the sake of readability, we have moved that discussion to the end of the Referee 2 section of the rebuttal.

We should address first the many relevant remarks raised by the Referee about the specific case of MIR driven, underdoped YBCO_{6.48}, which is the core topic of this paper.

In contrast, the present study involved chirping the 150 fs MIR pulse to a duration of ~1 ps by using a 10 mm NaCl rod, and observed the pump-induced magnetic signal only during the driving period of the MIR pump-pulse. This represents a departure from their prior work in the pump-probe time-delay.

The Referee is correct in noticing that the excitation protocol is different in this paper than in some of the previously reported studies on underdoped YBCO. In the original manuscript (ref. 17 in the main text) where light induced SC in YBCO was discovered, the material was excited with a 300 fs long pulse. As correctly recollected by the Referee, in these experiments SC-like optical features were observed over a time window of approximately 1-2 picoseconds after excitation.

However, in more recent studies, we have investigated the effect of different pump pulse durations on the dynamics (ref. 20 of the Main Text). We found that pumping with longer pulses creates a better and longer-lived SC-like state (see figure R2.1). In the present work, where the experimental signal is extremely small and each curve requires many days of accumulation, even factors of 2 in the size of the signal are crucial. For this reason, the experiment was performed with 1 ps pump pulses.

Representative results of reference 20 are displayed below in figure R2.1: (1) the lifetime of the transient superconducting-like properties grows linearly with the duration of the pump, and (2) 1 ps pump pulses (or longer), achieve the strongest SC-like state (fig. R2.1b), which saturate at the equilibrium value seen at ($T < T_c$).

To expand on this issue, part of the misunderstanding may be caused by a different definition of “lifetime”. In the early papers we reported the decay of the spectrally integrated THz response. In more recent work, and in ref. 20 we apply a far stricter definition of “superconductivity lifetime”. We only consider time delays where the imaginary conductivity diverges as $1/\omega$ and the transient superfluid density is finite. Once again, ref. 20 provides the modern definition of lifetime in view of better and more consistent datasets.

Figure R2.1: (a) Lifetime of the photoinduced superconducting state, estimated from the time window showing a finite value of the “superfluid density”, extracted from $\omega\sigma_2|_{\omega\rightarrow 0}$, as a function of pump pulse duration. (b) Peak value of the “superfluid density” as a function of pump pulse durations. These measurements were performed at a base temperature $T = 100\text{K}$, at a constant peak electric field of 2.5 MV/cm .

The current study appears to challenge or refute the possibility of light-induced superconductivity in their previous research as no effect is induced after pulse duration.

See above. The lifetime of the magnetic field expulsion reported in the present paper is of the same order as that observed for the superconducting optical features when pumping $\text{YBCO}_{6.48}$ with 1 ps long pump pulses (see figure R2.1a and ref. 20). Reference 20 is the only systematic study of the relaxation time as a function of pulse duration, and is the one that should be taken as relevant in this discussion.

Also, one should be careful when comparing time resolved changes in conductivity with time resolved magnetic field expulsion. Different observables can easily have slightly different temporal and spatial profiles, especially at these short timescales. This is evident in the data of figure 5, where delayed responses can be detected for different spatial positions of the probe.

To substantiate the reported light-induced phenomenon as superconductivity, it is imperative for the authors to replicate the pump conditions in prior research and observe transient diamagnetism within the same time-delay window, that is, 0.8 – 2 in YBCO/LBCO or over 20 ps in LESCO1/8 after pumping.

Each one of the time-resolved traces measured here takes more than one week, and we are not yet at the stage where systematic measurements can be done. Improvements of our experimental apparatus, already under way, will enable more comprehensive measurements in the future.

Here, we can only address the case of $\text{YBa}_2\text{Cu}_3\text{O}_{6.48}$ for which we have provided an extensive and conclusive body of evidence for photo-induced Meissner-like diamagnetism.

2. In the introduction and Figure 1, the authors illustrate that an 800 nm NIR pulse polarized within the ab-plane suppresses superconductivity below T_c , while an MIR pulse along the c-axis induces transient superconductivity above T_c . The authors seem to imply that the NIR

pulse may not induce transient superconductivity above T_c , and the MIR pulse may not suppress superconductivity below T_c . However, their previous research on LBCO (e.g., PRB 90, 100503R (2014)) showed that NIR excitation at 800 nm polarized along c-axis not only enhanced superconductivity below T_c but also induced transient superconductivity above T_c . In their research on YBCO (ref.19), the authors also showed that, in addition to the MIR pulse, NIR excitation can induce transient superconductivity. Specifically, in ref. 19, the authors found that an 800 nm NIR pulse on YBCO actually produced the largest condensate $\omega\sigma(2)|_{\omega\rightarrow 0}$, and a 400 nm pulse also induced sizeable $\omega\sigma(2)|_{\omega\rightarrow 0}$.

The statements above are not entirely accurate. As highlighted previously, whether a specific excitation geometry suppresses or enhances superconductivity depends on the details of the experiment, polarization, wavelength, doping and other details.

1) Femtosecond excitation of **optimally doped films of YBCO disrupts superconductivity**, when the pump polarization is **oriented in the ab planes**. This is **well documented and is universally accepted** (see ref. 25 of the main text for a comprehensive review).

2) When exciting along the c-axis the situation is different. As the Referee states, and as discussed above, in LBCO and in **underdoped YBCO**, 800 nm excitation with pulses **polarized along the c-axis** can induce a divergent $1/\omega$ response. The c-axis NIR excitation phenomenon (known to be associated with stripe melting in LBCO), is still to be understood for YBCO. The c-axis NIR excitation conditions are not studied here, and only (1) is relevant for this paper.

In the current study, the authors only demonstrate the effects of the 400 nm pump under the condition of $T < T_c$ and the 15um pump under the condition of $T > T_c$, as presented in Figures 3 and 4-6, respectively. Such comparisons are inappropriate. To validate their previous findings and ensure consistency, the authors should include the effects of both the NIR (or 400 nm) pump and MIR pump below and above T_c . To address the apparent controversies in different studies, they should also check the dependence of transient signal on pump pulse polarization, if possible.

The logic of the paper is misrepresented in the statement above. We are not comparing the results of figures 2 and 3 with those of figures 4-6.

The purpose of figures 2 and 3 is simply to validate the experimental technique. No discovery or new information is contained in figures 2 and 3. Simply, we consider a situation in which everybody agrees that superconductivity is being reduced (in-plane photo-excitation using 400 nm pulses), to validate the ultrafast magnetometry technique. Figures 2 and 3 were intended to provide the reader with confidence that ultrafast magnetometry can reliably track the Meissner state dynamics with sub-picosecond time resolution.

As requested by this Referee, we have now carried out 400-nm in plane pump experiments in optimally doped YBCO also above T_c . The experiments are performed in the same geometry as figure 3 of the main text (pump polarization in the ab-plane) tracking the evolution of the

Figure R2.2: Pump-induced changes in the measured magnetic field ΔB as function of pump-probe delay measured on top of the sample center, at a temperature $T = 100$ K, with an applied field of 2 mT.

magnetic field on top of the sample center where a reduction of the local magnetic field would be observed if photo-induced superconductivity appeared. In these measurements, we did not see any pump-induced magnetic field change (see figure R2.2).

Note also that c-axis excitation is not possible in this geometry. For these experiments (pump and probe impinging on the sample from opposite sides, see Fig. 3) we need thin film samples. However, thin films of sufficient quality can only be obtained with an ab surface, whereas films with the c-axis in the plane are (1) not easily found and (2) have not yet been used to study light induced enhancement of superconductivity.

Even if we decided to study the 400 nm excitation in bulk samples like those of figure 4, an entirely new design would be needed. In fact, the experiments of figure 4 rely on the use of Al_2O_3 to shield the region in which the magnetic field is measured from the pump (in no circumstance we want to spatially overlap pump beam and GaP detector). However, it is difficult to identify a material (like Al_2O_3 for the MIR pump) that would be able to shield the detector from the 400nm pump while letting the probe through.

We hope that the Referee will be sympathetic with the difficulties of these experimental geometries and our attempts to push the frontiers of ultrafast magnetometry in superconductors.

The Referee also requests measurements with MIR, c-axis excitation below T_c in the conditions of figure 4. We have collected data in this configuration at a series of temperatures down to $T = 45$ K $< T_c$. At $T = 45$ K $< T_c$, where the equilibrium state is already diamagnetic, we observe an enhanced magnetic field expulsion. **The results are encouraging and at first glance, they are consistent with the previously observed enhancements in superfluid density when pumping below T_c (ref. 18).**

However, the enhancement observed at 45 K should be interpreted with caution.

As shown in figure R2.4a, when one excites the normal state, one induces a superconducting-like thin layer above a non-superconducting (and weakly paramagnetic) bulk. This makes the measurements of figure 4 very “clean”, as the magnetic response of the thin photo-excited

layer dominate the overall evolution of the magnetic field outside the sample. This is not the case when exciting below T_c .

Indeed, below T_c we have a diamagnetic equilibrium superconductor beneath the photo-excited layer (see figure R2.4b). This is likely to affect the magnetic field dynamics after photo-excitation.

This makes it difficult to compare these two situations and for this reason we would prefer to not include these data in the manuscript.

In the future, we envisage experiments on thin lamellae of c-axis cut YBCO, which have a thickness comparable to the penetration depth of the pump. This measurement requires advanced fabrication methods (likely focused ion beam milling) which are beyond the scope of this work.

Figure R2.3: Pump-induced changes in the measured magnetic field (ΔB) as function of pump-probe delay measured at three different temperatures of 45 K (blue), 65 K (orange), and 100 K (red). The values are normalized to the 100 K peak value since these measurements are performed with a different detector (ZnTe (100)) compared to those reported in the main text.

Figure R2.4: Different boundary conditions for the experiments performed above and below T_c . **(a)** Above T_c , the photoexcited region (light green) is surrounded by YBCO in its normal state which shows a very weak magnetic response. **(b)** Below T_c the photoexcited region is surrounded by YBCO in its superconducting state featuring a strong diamagnetic response.

Following are comments and concerns on the measurement in this work:

1a) In the field of ultrahigh-power pulse laser technology, it is well established that irradiating a solid metallic target with a high-power pulse laser can generate a very high pulse magnetic field. This phenomenon is attributed to the generation and expelling of fast electrons from the laser focus spot due to the intense pulse laser driving, resulting in a high electrostatic potential near the focus and charging of the target. The potential then drags background cold electrons to the focus spot, leading to a current that generates a strong pulse magnetic field. Previous studies have demonstrated that a single turn coil with an open-ended plate target can produce a magnetic field of several hundred tesla or higher near the coil within the pulse laser driving period on target. See, for example, Zhu et al. APL 107, 261903 (2015), APL 113, 072405 (2018); Sederberg et al. PRX 10, 011063 (2020).) It is important to note that the induced electron current passing through the straight sample segments connecting the target can cause a pulse magnetic field even around the straight sample segments. In the present study, the mid-infrared (MIR) laser pulse has a peak field of approximately 2.5 MV/cm, raising the question of whether this could result in a magnetic field change of 10 μT during the pulse driving period. This effect has to be carefully considered.

The Referee is right, the pump alone can generate a large magnetic field in the sample. Not only, the magnetic field contained in the pump pulse ($\sim 1\text{T}$) is far larger than the one applied by our coil (10 mT) or expelled by a superconductor for mT scale applied fields.

The experiment is designed to exclude this phenomenon (see supplementary section S4). In short, the measurement is performed by **chopping the pump pulse (at 1kHz frequency) and alternating the polarity of the applied magnetic field (at 10 Hz frequency)**. Hence, the time dependent magnetic field measurement (Faraday rotation of the probe) is performed by subtracting the pump-induced response for applied $+B_{\text{ext}}$ from that measured for applied $-B_{\text{ext}}$. By doing so, **all contributions that do not depend on the applied magnetic field will cancel out**.

Figure R2.5: (from Figure 6, Main Text) Peak magnetic field expulsion measured for different applied magnetic fields at a fixed time delay $t = 0.75$ ps and at a base temperature $T = 100$ K. The error bars denote the standard error on the mean. The measured magnetic field expulsion is vanishingly small for zero applied magnetic field.

Because the mechanism proposed by the Referee would not depend on the externally applied magnetic field, it would not yield to a signal in our measurements.

This is further confirmed when looking at the scaling of the measured time dependent magnetic field expulsion with the applied external magnetic field, which is shown in Figure 6a and R2.5 below. **When no external magnetic field is applied, no magnetic field expulsion is measured.** If the magnetic field were generated by the pump, we would measure a signal also for the externally applied magnetic field $B_{\text{ext}} = 0$. We have added a paragraph in supplementary section S4 and in the main text that clarifies this.

2a) *The measurement may be affected by an additional complexity, as the MIR pump pulse itself contains a magnetic field component exceeding 0.8 T. Given that the lifetime of this transient signal closely matches the pump pulse duration, the authors must eliminate the possibility that the detected transient signal arises from the magnetic field component of the pump pulse propagating in the sample. 10 μT is several orders lower than the peak magnetic field of MIR pulse. Is a tiny misalignment of pump pulse can affect the measurement? Carefully designed experiments to investigate the interference of the pump pulse's magnetic field component on the experimental results seem to be necessary.*

The estimate made by the Referee of the magnetic field component in the pump pulse is correct ($> 0.8\text{T}$), that is many orders of magnitude larger than the dynamical magnetic field expulsion signal. We refer the Referee to the response above.

As an additional check, we have also confirmed that the measured magnetic field expulsion comes from the sample and not from the light pulse by observing that the measured signal vanishes not only for zero applied magnetic fields, but also in a number of additional measurements reported in Supplementary Section S10 (reported here in Figure R2.6 for convenience). We find that when the pump and probe beams are displaced laterally on the detector to a region where no sample is present, the signal also vanishes despite having both beams on the detector. (see Figure R2.6a).

Figure R2.6. (from Figure S10.1, Supplementary Information) (a) Sketch of the experimental geometry. The pump and the probe beams are both moved parallel to the edge of the detector from an area where the YBa₂Cu₃O_{6.48} crystal is present underneath the GaP detector layer to one where it is not. (b) Measured pump-induced changes in the local magnetic field at a temperature $T = 100\text{ K}$, time delay $t = 0.75\text{ ps}$, and peak electric field of 2.5 MV/cm . The solid line is a guide to the eye. The error bars denote the standard error on the mean.

This proves (1) that the effect does not originate from the pulse itself but rather from its interaction with the sample and (2) that it does not originate from a pump induced response in the detector assembly, but is given by the YBCO_{6,48} sample. We have added a paragraph in supplementary section S4 and S10 that underscores this.

3a) In the magneto-optic sampling of GaP (100) below T_c, a reduction in polarization rotation at the film's center and an increase near its edge were observed. The authors suggested that this change in polarization rotation is linked to a change in the magnetic field. However, scanning the probing pulse from the edge to the center not only alters the magnetic field experienced by the GaP layer but also changes the device's geometry. At the sample's edge, the probing light is reflected by the air-GaP-air multilayers, while at the center, the air-GaP-YBCO multilayer reflects the probing pulse. These different interfaces add complexity to extract a pure magnetic repulsion signal from the total recorded signal, and the authors should discuss possible additional effect from the GaP-YBCO interface.

This is true. We have also considered this effect. Let us to clarify this point better.

The experiment is designed specifically to mitigate these issues. In short, **three measures** ensure that a pure magnetic signal is extracted even when the probing position is moved.

First, the probe beam remains fixed in these scans while the sample is moved, meaning that no changes in incidence angle are produced as we probe different areas of the detector.

Second, the GaP detection crystal is wedged to ensure that only the reflection from the surface of the detector closer to the sample makes its way into the detection path.

Third, extra reflections from the YBCO are not picked up since the sample is mounted at a small angle (~ 1 degree) with respect to the GaP crystal. This angle is small enough not to interfere with the detection of the magnetic field expulsion but large enough to be able to spatially filter out the beam reflected from the YBCO interface. Figure R2.7 illustrates the experimental geometry highlighting the wedge angles and mounting angles. Wedge and YBCO tilt angles are exaggerated for clarity.

We have added this figure and further details about this in the description of the experimental geometry in Supplementary Information section 2.

Figure R2.7. Experimental geometry for the optical magnetometry measurements performed in the manuscript. The figure highlights the different paths taken by the reflections from different interfaces. Wedge and YBCO tilt angles are exaggerated for clarity. In reality $\alpha_{\text{wedge}} \sim 1.5^\circ$ and $\alpha_{\text{YBCO}} \sim 1^\circ$.

Furthermore, as explained in Supplementary Section S4 the polarization analysis is achieved by normalizing the measured difference in intensity between the s- and p- photodiode channels by their sum (overall intensity). Hence, changes of the reflectivity at the second GaP surface due to a change of medium between air and YBCO would not yield a change in the measured polarization rotation.

As further proof that our measurement is not affected by this change in the interface, we acquire probe-position-dependent scans across the YBCO₇ sample edge at a temperature $T > T_c$ (where the sample should not modify the applied field). We then compare it to one measured for $T < T_c$ (where a diamagnetic response should appear). The results of these measurements are shown in Figure R2.8.

Figure R2.8: (also in Figure S7.1b, Supplementary Information) Ratio of the local magnetic field B to the applied one B_{ext} , measured as function of distance across the straight edge of the $\text{YBa}_2\text{Cu}_3\text{O}_7$ half disc (see figure S2.1) for a temperature $T = 25\text{K} < T_c$ (blue line) and $T = 60\text{K} > T_c$ (gray line). The solid lines are guides to the eye.

Whilst the $T < T_c$ shows an enhanced field at the edge and an expulsion on top of the sample (arising from the sample diamagnetic response), the $T > T_c$ one (gray line) shows no changes in the measured magnetic field. These observations confirm that the measured signal is the pure magnetic expulsion signal from the YBCO sample and not an artefact coming from the GaP-YBCO/air interface. For completeness, we have improved the discussion in Supplementary Information section 4 and added this additional data in section 7.

Furthermore, a temperature-dependent measurement of the polarization rotation at the center and around the edge of the sample is obviously necessary. If the recorded polarization rotation signal inherently reflects the Meissner effect, the authors should observe a sudden change in the signal when the temperature is cooled below T_c , similar to the observation in the magnetic susceptibility measurement.

Following the Referee's suggestion we have extended our previous equilibrium temperature dependent measurements on the bulk $\text{YBCO}_{6.48}$ single crystal. The results are very encouraging.

For these equilibrium measurements the lock-in frequency is determined by how fast the magnetic field polarity can be inverted (10 Hz). Because of this low frequency some unwanted noise remains in the magnetic signal. To obtain better signal to noise ratio, we used a ~ 250 μm thick GaP detector to obtain larger polarization rotation signals. We focused on probing

the magnetic field expulsion above the sample rather than the edge to mitigate possible issues due to sample movements as the temperature is changed.

The results of these measurements are reported in figure R2.9a (orange filled symbols) where a sudden change in the signal is observed as the sample is cooled below T_c .

Figure R2.9b shows the magnetic susceptibility extracted from these data via the same magnetostatic calculations discussed in supplementary section S6 (red filled symbols). The extracted curve is compared to the DC magnetic susceptibility re-measured on the same sample using a commercial SQUID magnetometer (blue line). The agreement between the two curves is very good.

Figure R2.9: (also Figure S6.2, Supplementary Material) **(a)** Ratio of the local magnetic field B to the applied one B_{ext} , measured as function of temperature on top of the YBa₂Cu₃O_{6.48} crystal. **(b)** Magnetic susceptibility extracted via magnetostatic calculations from the optical magnetometry data compared to that measured on the same sample with SQUID magnetometry.

4a) In the context of c -axis measurements, it is important to address the lack of a spatial dependence scan at equilibrium, a procedure undertaken for the ab -plane device. Furthermore, conducting a temperature dependent scan at a specific distance would help clarify the connection between the recorded pump-probe signal and the superconductivity-induced Meissner effect.

We have acquired a spatial dependence scan at equilibrium for the YBCO_{6.48} single crystal sample at a fixed temperature $T = 25\text{K} < T_c$. The results of this scan are shown in figure R2.10. These measurements were performed using the same geometry, detector type and thickness of those shown throughout figures 2 to 5 in the main text, applying a 2 mT magnetic field that periodically switched polarity at a frequency of 10 Hz.

Similarly to the data collected for the YBCO₇ thin film and shown in figure 2 of the main text, we observe a reduction of magnetic field above the sample center and an enhancement near the edge. This confirms that the geometry of the experiment does not qualitatively affect our observations. Because our signal to noise ratio is reduced in the equilibrium measurements due to a lower lock-in frequency, we performed this spatial scan only at the lowest temperature, where the signal was the strongest.

In these data, the amount of field enhancement measured near the edge peaks at $\sim 2\%$, roughly one order of magnitude higher than what observed in the out-of-equilibrium measurements on the same sample.

Figure R2.10: Ratio of the local magnetic field B to the applied one B_{ext} , measured as function of distance from the edge of the $\text{YBa}_2\text{Cu}_3\text{O}_{6.48}$ single crystal. The measurement is performed at $T=25\text{K}$, applying an oscillating (10 Hz) magnetic field with 2mT peak amplitude. The solid line is a guide to the eye.

The quantitative difference between the equilibrium and the dynamical measurement is well understood based on the following considerations.

1) In our optical experiments, because the polarity of the magnetic field is switched periodically (see supplementary section S4) the equilibrium measurements performed below T_c are effectively analogous to a Zero Field Cooled (ZFC) measurement, as the amplitude of the applied field is changed while the sample is in the superconducting state.

In the dynamical measurements above T_c , the material is photo-excited in a field, and hence these are the equivalent to a Field Cooled (FC) type of measurement.

ZFC measurements generally give rise to a larger signal compared to FC measurement. For reference, in figure R2.11(a) we compare equilibrium ZFC and FC susceptibility measurements on the same $\text{YBCO}_{6.48}$ single crystal used for the experiment. These data highlight that even at temperatures well into the superconducting state, the FC magnetic susceptibility of $\text{YBCO}_{6.48}$ is at least 50 times smaller than the ZFC one.

Figure R2.11: (a) Temperature dependent DC magnetization measurements (ZFC: zero field cooled, FC: field cooled) highlighting the superconducting transition in $\text{YBa}_2\text{Cu}_3\text{O}_{6.48}$. The measurements were performed in a 1mT applied field perpendicular to the crystal c -axis. **(b)** Results of a magnetostatic calculation accounting for the geometry and placement of the detector that relate the sensed magnetic field change to a change in the magnetic susceptibility of the photo-excited region in $\text{YBa}_2\text{Cu}_3\text{O}_{6.48}$. The two points indicate the zero-field cooled and field cooled susceptibility of the $\text{YBa}_2\text{Cu}_3\text{O}_{6.48}$ at equilibrium.

To bring this into perspective we use the ZFC and FC magnetic susceptibility values to estimate the magnetic field that would be measured, using our technique, 50 μm away from the straight edge of a $\sim 150\ \mu\text{m}$ radius, $\sim 2\ \mu\text{m}$ thick half disc resembling the shape of the photo-excited region. As can be seen in Figure R2.11(b) the ZFC and FC susceptibility values yield magnetic fields outside the sample that are different by almost a factor of 1000, further underscoring the difficulty of quantitatively comparing above T_c and below T_c measurements.

2) Equilibrium and non-equilibrium expulsion are only qualitatively comparable (and not quantitatively). Even if ZFC and FC susceptibilities were equal, the effective geometries of the experiments performed in and out-of-equilibrium are different as shown in figure R2.12(a,b). Since the $\text{YBCO}_{6.48}$ sample is a bulk single crystal (2 mm x 0.5 mm, **2 mm thick**), when cooled below T_c the sample becomes superconducting throughout the whole volume. In the non-equilibrium case, on the other hand, we expect magnetic field expulsion to appear only within the photo-excited region, a half-disc ($\sim 150\ \mu\text{m}$ radius), with a thickness determined by the penetration depth of the pump pulse ($\sim 2\ \mu\text{m}$).

Figure R2.12: (a) Experiment geometry for equilibrium measurements below T_c . Here, the whole $\text{YBCO}_{6.48}$ crystal becomes superconducting. (b) Experimental geometry for out-of-equilibrium measurements above T_c . Only a small portion of the sample, of thickness comparable to the penetration depth of the pump ($\sim 2\ \mu\text{m}$), becomes superconducting. (c,d) Magnetic field profiles extracted as a function of distance from the sample edge using magnetostatic calculations and assuming a volume susceptibility $\chi_v = -0.3$. The calculations were performed for a bulk sample (c), representing the geometry of the equilibrium measurements, and a thin film (d), representing the geometry of the out-of-equilibrium ones.

This difference in geometry gives rise to different field profiles and as a consequence, different field strengths outside of the sample. To illustrate this, we compare in figure R2.12(c,d) the

calculated spatial dependence of the field in the vicinity of the sample edge calculated for $\chi_v = -0.3$ in the bulk geometry (equilibrium) and thin film geometry (out-of-equilibrium). Besides the different functional dependence, the value of the magnetic field enhancement outside the sample is sensitive to the specific geometry and is up to a factor of 3 stronger in the case of the bulk geometry, showing that even in an ideal case a one-to-one comparison is not possible.

Hence, whilst an enhancement is observed also in equilibrium below T_c , on the bulk YBCO_{6.48} sample, a quantitative comparison of the results between equilibrium and out-of-equilibrium measurements requires a number of estimates highlighted above. In the future, measurements on YBCO_{6.48} crystals micro-structured to resemble the shape of the photo-excited region and improved detection methods allowing equilibrium measurements in a field cooled configuration will provide means for a more direct comparison.

In summary, the ultrafast magneto-optic sampling method presented in this study could facilitate further investigation of the previously reported light-induced superconductivity. To validate this phenomenon, it is essential for the authors to reproduce the pump conditions used in earlier research and observe transient diamagnetism within the same time-delay window. Additionally, rigorous efforts are required to ascertain the intrinsic connection between the recorded pump-probe signal and the transient Meissner effect. All the above concerns should be adequately addressed.

As discussed above, a systematic study over many materials, which for the THz conductivity has required a decade of research, cannot be reproduced here in the far more challenging experimental conditions of ultrafast magnetometry.

We believe that for the **specific case of underdoped YBCO**, the results presented are well substantiated and unambiguously demonstrate that when the apical oxygen phonons are excited, the material acquires Meissner-like diamagnetism, which we feel deserve dissemination at in the most visible venue.

Lastly, we recommend that the authors include pertinent references in their paper to address the ongoing debate surrounding the light-induced superconductivity-like response in optical pump THz probe measurements.

We have also included pertinent references regarding the ongoing debate regarding optical-pump THz-probe measurements.

Here, we return to the issue introduced at the beginning of this report, and clarify the phenomenology and lifetimes observed in different high-T_c cuprates under different types of excitation:

LESCO. The original experiments in LESCO explored a different avenue for light enhanced superconductivity compared to YBa₂Cu₃O_{6.48}. The idea in the LESCO experiment was to use light to melt a competing stripe order that reduces superconductivity in equilibrium. It was posited that light may remove these stripes and allow for hidden superconductivity to appear. To melt the stripes, which reside in the ab plane, one has to pump along the a or b direction. This was specifically achieved when the wavelength of the pump was **tuned to the phonon resonance** (near 17 μm wavelength). Melting stripe order by this strategy was later shown to

happen in a related compound using femtosecond x-ray probing. (M. Först, et al., Phys. Rev. Lett. **112**, 157002 (2014)).

In LESCO the timescale for the melting and for the return to the non-superconducting state is then determined by (1) structural distortions and strain propagation, which are and phenomena that are limited by the speed of sound. Nucleation and growth may also be part of the recovery kinetics. Hence the superconducting response appears within several picoseconds and disappears in hundreds of picoseconds.

Indeed, when pumping perpendicular to the planes in LESCO, no effect was observed. However, in LESCO, the apical oxygen mode along the c axis is farther in the infrared ($> 20 \mu\text{m}$). The c-axis experiments were a sanity check, in which we showed that when no resonant melting of the stripes was possible (c-axis off resonance pumping), no hidden superconductivity could be activated.

LBCO. For the case of NIR excitation in *LBCO*, which was an accidental discovery, it was shown by our work and by other studies that c-axis pumping can destroy competing stripes (like in LESCO), whilst also minimizing heating by hot quasi-particles. Note that NIR studies are always confounded by the dynamics of hot carriers, which heat up the sample rapidly and are less specific than the MIR phonon pump experiments. Be as it may, it appears that stripe melting and activation of hidden superconductivity is also possible when pumping LBCO in the planes. There are yet a number of other details that the Referee may or may not be interested in, but we also found that in LBCO the effect can only be induced when the stripes are incommensurate and it becomes sharper and longer lived when a magnetic field is applied (see Nicoletti et al. Phys. Rev. Lett. **121**, 267003 (2018)). All these observations make LBCO a case in itself. As in equilibrium cuprate research, the details of the dynamical response become material specific.

YBCO. For the case of YBCO mentioned by the Referee, yet a third physical situation is at play. Here, no melting of a competing charge ordered state is achieved (there is only a very weak CDW and no clear competing charge order in $\text{YBCO}_{6.48}$). Here, excitation of the apical oxygen phonons was originally found to lead to superconducting like optical conductivities over a time window of approximately 1-2 picoseconds after excitation with a 300 fs pulse (see Ref. 17 in the main text).

As mentioned elsewhere in the body of the rebuttal, more recent studies have shown that the lifetime gets progressively longer for longer drive pulses, yielding also stronger and clearer SC-like optical features (see Ref. 20 in the main text).

Referee #3

The manuscript "Magnetic field expulsion in optically driven YBCO" by S. Fava et al. deals with the subject of "transient superconductivity", an issue which has attracted significant attention over the past few years. As is clearly outlined in the introduction, a number of experiments has revealed the emergence of new types of coherence when underdoped YBCO is driven along the *c*-direction with MIR optical pulses. In the induced non-equilibrium state the (imaginary part of the) optical conductivity shows a $\sigma_2 \sim 1/\omega$ behavior, compatible with a zero frequency pole in $\sigma_1(\omega)$ which is the hallmark of superconductivity. The idea is that the optical pulses coherently excite apical oxygen phonons that induce the superconducting-like behavior.

The question, if one can really deduce a transient superconducting state from the optical data has stimulated a lot of controversy. This concerns the measurement process itself (cf. e.g. J. S. Dodge, PRL 130, 146002 (2023)) but also the question if an "equilibrium quantity" as the superfluid stiffness, is a signature of superconductivity if this quantity is formally extracted from $\omega\sigma_2(\omega)$ in a non-equilibrium state. **In this regard, I consider the work presented in this manuscript as a significant step forward since authors take the Meissner effect as a proxy for superconductivity, i.e. the (complete) expulsion of a static magnetic field from the interior of a superconductor.**

In the experimental setup the magnetic field is measured via the polarization rotation by the Faraday effect of a linearly polarized probe pulse, realized by propagation through a GaP crystal in proximity to the sample. This is first demonstrated for a superconductor in equilibrium where authors observe a significant reduction of the field above the sample and, due to the expulsion, an increased magnetic field at the edge of the sample. It is also demonstrated that the opposite effect is observed when the SC state is disrupted by an ultraviolet laser pulse.

The central experiment discussed in the manuscript is sketched in Fig. 4 and concerns the measurement of the magnetic field expulsion for an underdoped YBCO sample which is excited along the *c*-direction and brought into a transient state where $\sigma_2(\omega)$ indicates possible induced SC properties. It is also shown that the signal is independent of the probe polarization angle and therefore originates from a Faraday and not e.g. the Pockels effect.

Moreover, it is demonstrated that the temperature dependence of the magnetic field expulsion and the non-equilibrium "superfluid density" are close to each other suggesting a common origin of both effects. The paper closes with a discussion of two possible explanations for the observed diamagnetic signal, either due to a real transient SC phase in part of the sample or the amplification of pre-existing diamagnetic currents in the pseudogap phase.

I consider this as a very interesting paper with a novel approach which advances our understanding of the transient state of coherently driven superconductors. The paper is well written and appendices provide sufficient supplementary material so that one can understand the experimental setup and the evaluation of the data.

We thank the Referee for the statements above.

I have only one major issue which concerns the consistency of the results related to the discussion of Figs. 4, 5. The evidence for the conclusion would be significantly strengthened if the authors also gave the equilibrium data below T_c for this sample (i.e. similar to what they discuss in Fig. 2) which would also help to estimate the non-equilibrium data above T_c .

This issue was also raised by Referee 2, and we concur that this is important. We have now performed below T_c measurements for the $\text{YBCO}_{6.48}$ single crystal sample. Figure R3.1 shows the results of a spatial dependence scan measured at equilibrium at a fixed temperature $T = 25\text{K} < T_c$. These measurements were performed using the same geometry, detector type and thickness of those shown throughout figures 2 to 5 in the main text, applying a 2 mT magnetic field that periodically switched polarity at a frequency of 10 Hz.

Figure R3.1: Ratio of the local magnetic field B to the applied one B_{ext} , measured as function of distance from the edge of the $\text{YBa}_2\text{Cu}_3\text{O}_{6.48}$ single crystal. The measurement is performed at $T=25\text{K}$, applying an oscillating (10 Hz) magnetic field with 2mT peak amplitude. The solid line is a guide to the eye.

Similarly to the data collected for the YBCO_7 thin film and shown in figure 2 of the main text, we observe a reduction of magnetic field on top of the sample center and an enhancement near the edge confirming that the geometry of the experiment does not qualitatively affect our observations. In these data, the amount of field enhancement measured near the edge peaks at $\sim 2\%$, roughly one order of magnitude higher than what observed in the out-of-equilibrium measurements on the same sample.

The quantitative difference between the equilibrium and the dynamical measurement is well understood based on the following considerations.

1) In our optical experiments, because the polarity of the magnetic field is switched periodically (see supplementary section S4) the equilibrium measurements performed below T_c are effectively analogous to a Zero Field Cooled (ZFC) measurement, as the amplitude of the applied field is changed while the sample is in the superconducting state. In the dynamical measurements above T_c , the material is photo-excited in a field, and hence these are the equivalent to a Field Cooled (FC) type of measurement. ZFC measurements generally give rise to a larger signal compared to FC measurements. For reference, in figure R3.2(a) we compare equilibrium ZFC and FC susceptibility measurements on the same $\text{YBCO}_{6.48}$ single crystal used for the experiment. These data highlight that even at temperatures well into the superconducting state, the FC magnetic susceptibility of $\text{YBCO}_{6.48}$ is at least 50 times smaller than the ZFC one.

To bring this into perspective we use the ZFC and FC magnetic susceptibility values to estimate the magnetic field that would be measured, using our technique, $50\ \mu\text{m}$ away from the straight edge of a $\sim 150\ \mu\text{m}$ radius, $\sim 2\ \mu\text{m}$ thick half disc resembling the shape of the photo-excited region. As can be seen in Figure R3.2(b) the ZFC and FC susceptibility values yields

magnetic fields outside the sample that are different by almost a factor of 1000, further underscoring the difficulty of quantitatively comparing above T_c and below T_c measurements.

Figure R3.2: (a) Temperature dependent DC magnetization measurements (ZFC: zero field cooled, FC: field cooled) highlighting the superconducting transition in $\text{YBa}_2\text{Cu}_3\text{O}_{6.48}$. The measurements were performed in a 1mT applied field perpendicular to the crystal c-axis. (b) Results of a magnetostatic calculation accounting for the geometry and placement of the detector that relate the sensed magnetic field change to a change in the magnetic susceptibility of the photo-excited region in $\text{YBa}_2\text{Cu}_3\text{O}_{6.48}$. The two points indicate the zero-field cooled and field cooled susceptibility of the $\text{YBa}_2\text{Cu}_3\text{O}_{6.48}$ at equilibrium.

2) Equilibrium and non-equilibrium expulsion are only qualitatively comparable (and not quantitatively). Even if ZFC and FC susceptibilities were equal, the effective geometries of the experiments performed in and out-of-equilibrium are different. Since the $\text{YBCO}_{6.48}$ sample is a bulk single crystal (2 mm x 0.5 mm, **2 mm thick**), when cooled below T_c the sample becomes superconducting throughout the whole volume.

In the non-equilibrium case, on the other hand, we expect magnetic field expulsion to appear only within the photo-excited region, a half-disc ($\sim 150 \mu\text{m}$ radius), with a thickness determined by the penetration depth of the pump pulse ($\sim 2 \mu\text{m}$). This difference in geometry gives rise to different demagnetizing factors and as a consequence, different field strengths outside of the sample. To illustrate this, we compare in figure R3.3(a,b) the calculated spatial dependence of the field in the vicinity of the sample edge calculated for $\chi_v = -0.3$ in the bulk geometry (equilibrium) and thin film geometry (out-of-equilibrium). Besides the different functional dependence, the value of the magnetic field $50 \mu\text{m}$ away from the straight edge is sensitive to the specific geometry and is up to a factor of 3 stronger in the case of the bulk geometry, showing that even in an ideal case a one-to-one comparison is not possible.

Figure R3.3: (a) Experiment geometry for equilibrium measurements below T_c . Here, the whole $\text{YBCO}_{6.48}$ crystal becomes superconducting. (b) Experimental geometry for out-of-equilibrium measurements above T_c . Only a small portion of the sample, of thickness comparable to the penetration depth of the pump ($\sim 2 \mu\text{m}$), becomes superconducting. (c,d) Magnetic field profiles extracted as a function of distance from the sample edge using magnetostatic calculations and assuming a volume susceptibility $\chi_v = -0.3$. The calculations were performed for a bulk sample (c), representing the geometry of the equilibrium measurements, and a thin film (d), representing the geometry of the out-of-equilibrium ones.

Hence, while an enhancement is observed also in equilibrium below T_c , on the bulk $\text{YBCO}_{6.48}$ sample, a quantitative comparison of the results between equilibrium and out-of-equilibrium measurements requires a number of estimates highlighted above.

In the future, measurements on $\text{YBCO}_{6.48}$ crystals micro structured to resemble the shape of the photo-excited region and improved detection methods allowing equilibrium measurements in a field cooled configuration will provide means for a more direct comparison.

In the present version this is accomplished via a quasistatic calculation, which for the equilibrium case would be also more reliable.

The validity of our magnetostatic calculations is underscored by a series of verifications based on additional experimental measurements. In figure R3.4 we compare the temperature

dependent ZFC magnetic susceptibility measured with a commercial SQUID magnetometer to that extracted from the magnetic field values measured using optical magnetometry using the same magnetostatic calculation described in supplementary section S6. The agreement between the two curves is very good, providing a validation of our method.

Figure R3.4: (also Figure S6.2, Supplementary Information) **(a)** Ratio of the local magnetic field B to the applied one B_{ext} , measured as function of temperature on top of the $\text{YBa}_2\text{Cu}_3\text{O}_{6.48}$ crystal. **(b)** Magnetic susceptibility extracted via magnetostatic calculations from the optical magnetometry data compared to that measured on the same sample with SQUID magnetometry.

Moreover, in previous work of the authors it has been shown that also below T_c , $\omega\sigma_2(\omega)$ is enhanced by the coherent excitation. If the interpretation of the authors is consistent this should also induce a concomitant enhanced magnetic field expulsion.

Regarding the effect of MIR excitation below T_c , we have also collected data in the same geometry as Figure 4 but at lower temperatures and also at $T = 45 \text{ K} < T_c$. The extended temperature dependence is shown in figure R3.3. One can observe that the effect saturates below 100K. At $T = 45 \text{ K} < T_c$ we observe a stronger magnetic field expulsion which is broadly compatible with previous results from MIR pump, THz probe experiments revealing an enhancement of the superconducting properties also below T_c .

Nevertheless, these results should be taken with caution. The observation of enhanced magnetic field expulsion below T_c is solid. However, the exact amount of field expulsion cannot be straightforwardly compared to that measured above T_c . In the above T_c measurements, the effects of the non-excited $\text{YBCO}_{6.48}$ bulk below the photo-excited region can be neglected (see figure R3.4a), as the normal state of $\text{YBCO}_{6.48}$ is nearly non-magnetic (weakly paramagnetic). However, below T_c this is not the case. The large diamagnetic properties of the equilibrium superconductor beneath the photo-excited layer (see figure R3.4b) may affect the magnetic field dynamics in the photo-excited region. This makes it difficult to compare the degree of measured field expulsion measured above and below T_c . Due to these ambiguities, we decided to not pursue $T < T_c$ measurements further and would prefer to not include these data in the current version of the manuscript.

In the future, we envision that it will be possible to perform experiments on thin lamellae having a thickness comparable to the penetration depth of the pump. Although this will

ensure that the whole sample will be photo-excited, and will allow for an accurate determination of the degree of photo-induced magnetic field expulsion also below T_c , the measurement require advance fabrication methods (likely focused ion beam milling) which are beyond the scope of this work.

Figure R3.3: Pump-induced changes in the measured magnetic field (ΔB) as function of pump-probe delay measured at three different temperatures of 45 K (blue), 65 K (orange), and 100 K (red). The values are normalized to the 100 K peak value since these measurements are performed with a different detector (ZnTe (100)) compared to those reported in the main text.

Figure R3.4: Different boundary conditions for the experiments performed above and below T_c . **(a)** Above T_c , the photoexcited region (light green) is surrounded by YBCO in its normal state which shows a very weak magnetic response. **(b)** Below T_c the photoexcited region is surrounded by YBCO in its superconducting state featuring a strong diamagnetic response.

Provided, authors provide this additional material, I recommend publication of the manuscript in Nature Materials.

We thank the Referee for the positive assessment of the manuscript. We are confident that they will be satisfied with the additional material we provided and will recommend publication of the revised version of the manuscript.

Reviewer Reports on the First Revision:

Referees' comments:

Referee #1 (Remarks to the Author):

I read carefully the revised manuscript and the rebuttal, and basically I can agree with the authors on most parts. I believe that the work presented in this paper, if proven to be valid, will greatly advance the study and understanding of light-induced superconducting-like phenomena in cuprates.

I have one remaining suggestion for the authors. In response to the referee 2's concern regarding the inconsistency of reported light-induced superconductivity time windows, the authors argued that part of the misunderstanding might be caused by a different definition of superconductivity lifetime that the authors adopted in several of their prior papers. While this information was clarified in the rebuttal letter, it was not included in the revised manuscript. To avoid potential confusion from a wider audience, it is recommended to briefly clarify this definition somewhere within a revised manuscript.

One typo to correct: on page 4 of the main text, the GaP crystal thickness was stated as 75 μm , but in Figure 2, it was shown as 70 μm .

Referee #2 (Remarks to the Author):

I have thoroughly reviewed the authors' detailed response to the referees' comments and the revised manuscript. I appreciate the effort that the authors have made in addressing all the concerns raised in the referees' reports. Of particular significance is the inclusion of measurement data depicting pump-induced magnetic field changes below T_c in their response. It is essential to incorporate these data, specifically Figure R2.3 (or Figure R3.3), either in the main manuscript or as supplementary material. As highlighted in previous reviews, presenting pump-induced changes both above and below T_c is crucial for reader comprehension. I recommend the publication of the manuscript in Nature Materials contingent upon the inclusion of this additional information.

A minor issue:

The authors are advised to review the data acquisition formula (Δ_{θ} either pump_on or pump_off) provided in section S4 (page 9 in the supplementary file). It appears that the presence of the denominator (sum of signal) is unwarranted; its inclusion would prevent the cancellation of the magnetic field arising from either the high magnetic component of the pump pulse or the field induced possibly by the movement of background cold electrons due to the intense pulse driving.

Referee #3 (Remarks to the Author):

The authors have convincingly replied to the comments I have raised in my previous report. In particular, I appreciate that the expulsion in the equilibrium SC state of YBCO_{6.48} is now discussed in the SM.

I therefore recommend publication of the manuscript in Nature Materials.

Detailed Reply To Referees - Nature 2023-12-22464A

In this document comments from the Referees are included in **black**. Author responses are displayed in **blue**.

Referees' comments:

Referee #1 (Remarks to the Author):

I read carefully the revised manuscript and the rebuttal, and basically I can agree with the authors on most parts. I believe that the work presented in this paper, if proven to be valid, will greatly advance the study and understanding of light-induced superconducting-like phenomena in cuprates.

I have one remaining suggestion for the authors. In response to the referee 2's concern regarding the inconsistency of reported light-induced superconductivity time windows, the authors argued that part of the misunderstanding might be caused by a different definition of superconductivity lifetime that the authors adopted in several of their prior papers. While this information was clarified in the rebuttal letter, it was not included in the revised manuscript. To avoid potential confusion from a wider audience, it is recommended to briefly clarify this definition somewhere within a revised manuscript.

Following the Referee's suggestion we now clarify the new definition of lifetime in the discussion of the THz results which appears in the legend of Extended Data Figure 1.

One typo to correct: on page 4 of the main text, the GaP crystal thickness was stated as 75 um, but in Figure 2, it was shown as 70 um.

We thank the Referee for highlighting this inconsistency. We have corrected the labels in the figure according to the Main Text.

Referee #2 (Remarks to the Author):

I have thoroughly reviewed the authors' detailed response to the referees' comments and the revised manuscript. I appreciate the effort that the authors have made in addressing all the concerns raised in the referees' reports. Of particular significance is the inclusion of measurement data depicting pump-induced magnetic field changes below T_c in their response. It is essential to incorporate these data, specifically Figure R2.3 (or Figure R3.3), either in the main manuscript or as supplementary material. As highlighted in previous reviews, presenting pump-induced changes both above and below T_c is crucial for reader comprehension.

Following the Referee's suggestion we now include pump-induced changes measured below \$T_c\$ in Supplementary Information S5.

I recommend the publication of the manuscript in Nature Materials contingent upon the inclusion of this additional information.

We thank the Referee for their careful reading of our manuscript and for recommending publication of the revised manuscript.

A minor issue:

The authors are advised to review the data acquisition formula ($\Delta\theta$ either pump_on or pump_off) provided in section S4 (page 9 in the supplementary file). It appears that the presence of the denominator (sum of signal) is unwarranted; its inclusion would prevent the cancellation of the magnetic field arising from either the high magnetic component of the pump pulse or the field induced possibly by the movement of background cold electrons due to the intense pulse driving.

We thank the Referee for spotting this issue in our notation. Indeed the $\Delta\vartheta_{\text{pump-off}}$ and $\Delta\vartheta_{\text{pump-on}}$ signals are not divided by the sum of the positive and negative magnetic field components. We have corrected the definition in the revised version of the Supplementary Information.

Referee #3 (Remarks to the Author):

The authors have convincingly replied to the comments I have raised in my previous report. In particular, I appreciate that the expulsion in the equilibrium SC state of YBCO_6.48 is now discussed in the SM.

I therefore recommend publication of the manuscript in Nature Materials.

We thank the Referee for their positive criticism and for recommending publication of the revised manuscript.